



# Very-high resolution aerial imagery and deep learning uncover the fine-scale spatial patterns of elevational treelines

Carrieri Erik[1], Morresi Donato[1], Meloni Fabio[1], Anselmetto Nicolò[1], Lingua Emanuele[2], Marzano Raffaella[1], Urbinati Carlo[3], Vitali Alessandro[3], Garbarino Matteo[1]

[1]Department of Agricultural, Forest and Food Sciences, University of Turin, Grugliasco, 10095, Italy
[2]Dept. of Land, Environment, Agriculture, University of Padova, Legnaro, 35020 ,Italy
[3]Dept. of Crop, Food and Environmental Sciences, Marche Polytechnic University, Ancona, 60131, Italy

*Correspondence to*: Erik Carrieri (erik.carrieri@unito.it)

**Abstract.** Treelines are sensitive indicators of global change, as their position, composition and pattern directly respond to numerous ecological and anthropogenic factors. Most studies are case-specific and treeline features vary greatly worldwide making it very difficult to model an overall pattern. Therefore, the further development of methods to accurately map fine-scale treeline spatial patterns, especially through innovative approaches such as remote sensing with unmanned aerial vehicles (UAV) and deep learning models, is of scientific importance for the conservation of forest ecosystems in the face of ongoing and future ecological challenges.

In this study, we aimed to fill this gap by combining field and UAV-based data with a deep learning model to retrieve single tree-scale information over 90 ha distributed on 10 study sites in the Italian Alps. Using the proposed methodology, we were able to correctly detect individual tree crowns of conifers taller than 50 cm with a detection rate of 70% and an F1 score of 0.76. The detection rates of individual tree crowns improved with increasing tree height, reaching a peak value of 86% when only tall trees (>2 metres) were considered. Canopy delineation was good when all trees were considered (Intersection over Union (IoU) = 0.76) and excellent when only tall trees were considered (IoU = 0.85). The estimates of tree position and height achieved an RMSE of 59 cm and 92 cm, respectively. Our univariate and bivariate heterogeneous Poisson Point Pattern Analysis (PPA) revealed a clustered pattern for spatial scales < 20 m, and a strong repulsion between small and tall trees at all the tested spatial scales, respectively. PPA results suggest that in the Alps, seedlings tend to progressively occupy safe sites and colonise non-competitive sites, resulting in the evenly sized clusters found. We demonstrated that the proposed methodology effectively detects, delineates, georeferences and, measures tree height of most trees across diverse Alpine treeline ecotones. This enables the analysis of fine-scale spatial patterns and underlying ecological processes. The inclusion of heterogeneous study areas facilitates the transferability of the segmentation model to other mountain regions and makes the present study a benchmark for creating a global network of fine-scale mapped treeline spatial patterns to monitor the effects of global change on ecotone dynamics.



## 1 Introduction

The elevational treeline is the transition zone from the uppermost closed montane forest (timberline) to the highest scattered trees (tree species line) (Holtmeier et al., 2003), and one of the most studied ecotones. Since the late 19th century scientific studies largely focused on the diversity and complexity of factors affecting the ecotone spatial and temporal patterns at different scales (Hansson et al., 2021; Holtmeier, 2009). It is well known that temperature plays a crucial role in treeline positioning and dynamics from regional to global scales (Dirnböck et al., 2003; Gehrig-Fasel et al., 2007; Harsch et al., 2009), but is not the only driving factor. Many other studies have emphasised the significant role of other factors in treeline formation (Mienna et al., 2024), including water availability (Barros et al., 2017; Williams et al., 2013), site topography (Leonelli et al., 2016; Marquis et al., 2021; Müller et al., 2016), biotic drivers (Brown and Vellend, 2014; Cairns et al., 2007) and anthropogenic pressure (Gehrig-Fasel et al., 2007; Malandra et al., 2019; Vitali et al., 2019).

Global change can trigger large-scale vegetation dynamics affecting the provision of ecosystem services - such as carbon sequestration (Mienna et al., 2024). Climate alteration can induce upward migration of species threatening a loss of habitat and biodiversity of high alpine communities habitat (Kyriazopoulos et al., 2017). This sensitivity to climatic and anthropogenic factors makes high-elevation ecotones key indicators of global change (Dirnböck et al., 2011; Greenwood and Jump, 2014). Monitoring changes at elevational treelines is therefore of utmost importance to follow how forests are responding and to forecast how they will respond to a changing environment (Chan et al., 2024; Hansson et al., 2023; Mottl et al., 2021) and ultimately to guide the definition of appropriate conservation strategies. However, the complex interaction of the above-mentioned drivers requires very heterogeneous systems capable to appreciate within wide spatio-temporal gradients soil and vegetation features over short distances (Holtmeier and Broll, 2007, 2017).

An open question in many fields of ecology is how to infer processes by observing patterns. In this context, the great spatial heterogeneity of this high-altitude ecotone hinders the transferability of case studies observations and therefore their generalization. How to tackle the spatial heterogeneity issue is still an open question, and consequently the attribution of the observed processes to specific drivers is still a challenge (Garbarino et al., 2023). Combining ground-based and remote sensing data could be a winning venue to solve this compelling issue, especially if pursued with a flexible and efficient protocol. Field surveys remain the traditional methods used also at treelines and involve measuring several tree parameters (e.g. stem DBH, height, position, health conditions) within small study areas – plots or transects (Mainali et al., 2020; Van Bogaert et al., 2011; Vitali et al., 2017, 2019). This approach supplies precise data, but is time-consuming. In addition to the limited spatial extension of the plots/transects, their lack of spatial contiguity can make them scarcely representative and the obtained results unsuitable to detect tree spatial distribution.

At this point remote sensing (RS) techniques come into play. The use of remote sensing techniques dates back to the 1980s (Holmgren and Thuresson, 1998), but it is only in the last two decades that they have been widely used in treeline ecology (Garbarino et al., 2023). The choice of the right RS tool involves the scientific question of which spatial and temporal scale we need to use. For instance, while satellite imagery can provide suitable data over large forest areas and wide time intervals



(Garbarino et al., 2020; Nguyen et al., 2024), most optical sensors do not provide the spatial resolution required for individual
tree mapping (Bennett et al., 2024; Morley et al., 2018; Simard et al., 2011). Within the limitations of field surveys (limited
spatial and temporal extent) and satellite-based data (high spatial and temporal extent and low resolution) that Uncrewed Aerial
Vehicle (UAV) platforms can bridge the gap and be applied in tree mapping (Fromm et al., 2019; Qin et al., 2022; Xie et al.,
2024). The increasing accessibility and friendly use of these aerial platforms make them increasingly valuable and efficient
for such applications. In addition to wall-to-wall mapping of relatively large and heterogeneous areas, drone survey enables
the analysis of fine-scale drivers and the extraction of tree attributes and features (Nasiri et al., 2021; Panagiotidis et al., 2017;
Shimizu et al., 2022; Xiang et al., 2024).
A single-tree scale approach is fundamental in treeline ecology, as it provides a better understanding of the underlying
ecological processes (i.e., drivers) essential for linking treeline dynamics to tree-to-tree interactions and facilitation-
competition relationships (Looney et al., 2018; Trogisch et al., 2021; Wang et al., 2021).
Convolutional Neural Networks (CNNs) based on very-high-resolution images are a reliable and versatile tool for single-tree
scale analyses, enabling the accurate identification and representation of different plant species and communities as well as
the detection of individual trees (Braga et al., 2020; Fricker et al., 2019; Fromm et al., 2019; Kattenborn et al., 2021). The
latter can be achieved through instance segmentation algorithms that enable the detection of individual objects on the input
images, allowing to distinguish and separate individual neighbour tree canopies (Ball et al., 2023; Braga et al., 2020).
Our general hypothesis is that RGB imagery derived from low-cost UAVs can provide data for accurate single tree detection
and extraction of attributes at the treeline ecotone. In particular, we expect tree height to have a strong influence on the
performance of the model, with better results for larger trees. Furthermore, we set the hypothesis that by training the DL model
with very different treelines images, we could improve the transferability of the model to different mountainous regions without
a significant drop in performance.
Despite the widespread use of UAV for single tree mapping and tree features detection in different forest ecosystems
(Dietenberger et al., 2023; Diez et al., 2021; Weinstein et al., 2019), a framework for mapping fine-scale tree spatial patterns
at treeline ecotones based on low-cost UAV imagery is still lacking. At this regard, the present study aims to: (i) propose a
methodology that combines very high-resolution RGB images derived from low-cost UAVs with CNNs to provide a single-
tree fine-scale perspective and infer processes from patterns; (ii) test  if the model is transferable to heterogeneous datasets and
validate the performance in single tree detection, crown delineation and estimating position and height; (iii) use the fine-scale
treeline maps output to analyse spatial patterns and investigate tree-to-tree interactions with the aim of testing the applicability
of our results to treeline ecology.



## 2 Materials and Methods

### 2.1 Study Area

We selected ten study sites across the Italian Alps (Fig. 1) covering a large longitudinal gradient to obtain a balanced dataset representative of the Western, Central, and Eastern Italian Alps, therefore showing highly heterogeneous climatic, topographical, soil, and vegetational conditions (Appendix A). For instance: (i) climate conditions vary from Atlantic to continental from West to East with influences of cold polar air descending from northern Europe and warm Mediterranean air flowing northward from the south. Mean annual temperatures range from 0° to 10° C and annual precipitation varies between 400 and 3000 mm with extremes in both variables strongly related to physiography (Isotta et al., 2014); (ii) topography consists of extensive lowlands, steep valleys and mountain peaks rising above 4800 m a.s.l.; (iii) above the closed forest the soil includes mesic and xeric regions, displaying a sequence of grasslands, sparsely vegetated areas, screes and surfaces strongly affected by gravitational events forming rill and gullies; and (iv) all the landscapes experienced centuries of humans land use practices with different degrees, yet land abandonment is more marked in the Western sector of the study area (Anselmetto et al., 2024; Bätzing et al., 1996). The introduction of such heterogeneity in the dataset was aimed at testing the transferability of the protocol to several treeline conditions. In line with the typical species coniferous composition of the subalpine belt in the Alps, in all the studied treelines the dominant treeline-forming species are European larch (*Larix decidua* Mill.) and Swiss stone pine (*Pinus cembra* L.). The study sites host also Norway spruce (*Picea abies* (L.) H.Karst.), dwarf mountain pine (*Pinus mugo* Turra), mountain pine (*Pinus uncinata* Miller), Scots pine (*Pinus sylvestris* L.) and a smaller presence of broadleaf species such as green alder (*Alnus viridi*s (Ehrh.) K. Koch) and silver birch (*Betula pendula* Roth). Further and more detailed information of the study sites are reported in Table 1.



**Figure 1.** Geographic location of (a) the Alpine Convention Perimeter in Europe and (b) the ten study sites (brown diamonds) along with their names across the Italian Alps. Detail in the UAV-derived orthomosaic of the study site (c) Devero and (d) same site overlayed with the CHM. (e) further details of the study area Devero and (f) its CHM. For further details see Sect. 2.2



**Table 1. Details of the study sites including date of the survey, their latitude and longitude (WGS84), average elevation (m a.s.l.), aspect, dominant tree species, mean annual temperature (°C) and total annual precipitation (mm). Climate variables were derived from Chelsa Climate database (Karger et al. 2020), while position, elevation, and species from the field surveys.**

| Study site | date | Latitude (°) | Longitude (°) | Elevation (m a.s.l.) | Aspect | Species | Mean annual temperature (°C) | Annual precipitation (mm) |
|---|---|---|---|---|---|---|---|---|
| Avic | 06/10/2021 | 45.697 | 7.593 | 2,184 | SE | *L. decidua, P. abies, P. uncinata* | 1.9 | 1115 |
| Becco | 28/09/2021 | 46.471 | 12.118 | 2,190 | N-NE | *P. cembra, L. decidua, P. abies* | 0.9 | 1449 |
| Bocche | 06/07/2021 | 46.338 | 11.744 | 2,245 | SW | *P. cembra, L. decidua, P. abies* | 0.7 | 1225 |
| Chianale | 29/06/2021 | 44.646 | 6.975 | 2,283 | N | *L. decidua, P. cembra* | 1.6 | 829 |
| Devero | 14/06/2021 | 46.316 | 8.294 | 2,186 | NW | *L. decidua* | 1.4 | 1631 |
| Genevris | 26/07/2021 | 45.030 | 6.897 | 2,379 | W | *L. decidua, P. cembra* | 1.4 | 1263 |
| Livigno | 22/07/2021 | 46.516 | 10.142 | 2,322 | NW | *L. decidua, P. cembra, P. mugo* | 0.1 | 1067 |
| Rion | 22/09/2021 | 45.830 | 7.262 | 2,290 | S-SE | *L. decidua, P. abies* | 0.7 | 1759 |
| Senales | 07/07/2021 | 46.727 | 10.898 | 2,319 | S | *L. decidua, P. cembra, P. abies* | 0.2 | 923 |
| Valfurva | 21/07/2021 | 46.454 | 10.461 | 2,371 | E | *L. decidua, P. abies, P. cembra* | 1.2 | 894 |

## 2.2 Sampling design and data collection

We randomly selected 10 treeline ecotones above 2,000 m a.s.l. along an eastern-western gradient in the Italian Alps with a minimum distance of 25 km. In these ecotones, we placed ten 9-ha square plots (300 m x 300 m) with a side parallel to the main slope so that the forestline occurred in the lower third of the plot. We defined forestline as the continuous line separating the closed forest (canopy cover > 10%) from the semi-open and open areas (canopy cover < 10%) (FAO, 1998). The canopy cover was assessed based on the pan-European Tree Cover Density (TCD) layer provided by Copernicus (https://land.copernicus.eu/en).

Data collection included UAV and field surveys in summer 2021. In particular, we employed a DJI Phantom 4 pro V2 quadcopter equipped with a RGB camera with a 1-inch CMOS sensor with 20 MP. UAV survey consisted of three flights: two of them with the camera in the nadiral position (one along the contour lines and one perpendicular to them), and the last one with an oblique camera perspective of 60° off-nadir, granting a more complete view of trees and terrain features. Image acquisition was achieved by performing swipes at a flight height of 30 m above the highest point of the 300m x 300m plot. To



avoid a drop in the spatial resolution in the bottom area of the plot due to the steepness, the three flights were repeated starting
from the central area of the plot, at approximately 150 m from the plot side. All the flights were performed on sunny, windless
days to avoid cloud coverage and to minimise image distortions due to UAV movements. Images frontal and lateral overlaps
were, respectively, 80 and 80% to ensure a comprehensive coverage of the surface. Prior to the UAV flights, 12 Ground
Control Points (GCPs) in the form of bull's eye targets were deployed in the imaged area and their position was assessed using
a Trimble R2 and Reach RS2 GNSS (Global Navigation Satellite Systems) antennas, both with sub-metric static horizontal
and vertical positioning accuracies with a 10-min occupancy. GCPs position was ultimately post-processed for a final
georeferencing correction. The acquired RGB aerial images were processed using Agisoft Metashape Pro software version
1.5.1. A Structure-from-Motion procedure was used to produce 3D point clouds which enabled the production of DSMs, and
5-cm spatial resolution orthomosaics. A ground classification from the aerial data point cloud was used to normalise the point
cloud and generate a digital terrain model (DTM). The DTMs and DSMs were used to extract the CHMs by computing the
above-ground height, thus enabling the discernment of ground and non-ground points in the respective point clouds. We
recorded position, height, and species of 50 randomly selected trees for each study site scattered across the plot. Tree's height
was assessed through a TruPulse 200b (Crisel srl) or measuring tape for smaller individuals. The position of the trees was
measured through the use of the above mentioned antennas with a 3- to 5-minute occupancy. The final ground control dataset
included a total of 500 trees.
The entire workflow of the study, from data acquisition to final analyses, is reported in Figure 2.





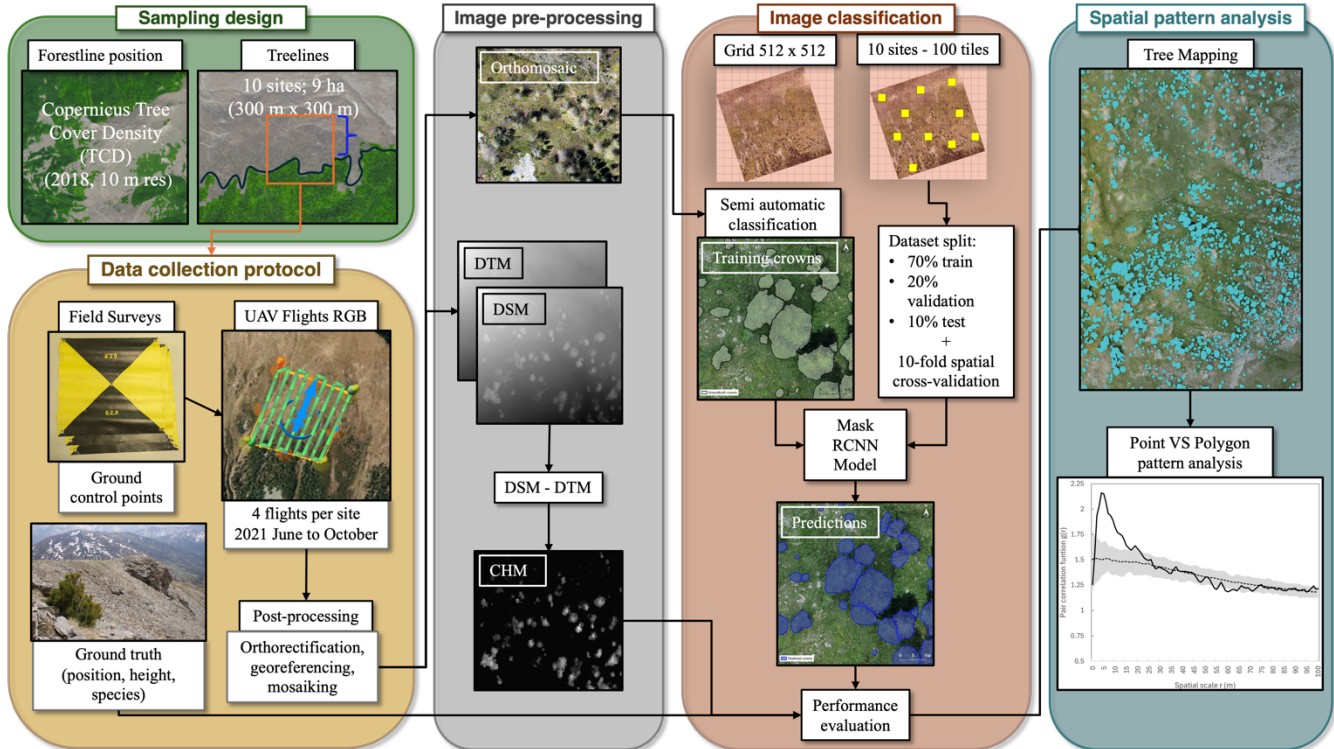


**Figure 2. Overview of the workflow adopted to conduct tree-scale analyses at the alpine treeline ecotone. Each box depicts a different**
**methodological step of the study.**

## 2.3 Deep learning modelling

To perform tree detection and segmentation we used a pre-trained deep learning (DL) model based on the Mask R-CNN
algorithm implemented in the "Detectron2" library from Meta AI and available at
https://github.com/facebookresearch/detectron2. Mask R-CNN is a DL framework which performs instance segmentation by
combining semantic segmentation and object detection (Kattenborn et al., 2021). Its framework involves the generation of
region of interest proposals by a deep fully convolutional network, and then there is a classification of the object of interest
within each generated region proposal. Our methodology consisted in the following steps: i) cropping RGB orthomosaic of
each study site into adjacent tiles of 512 x 512 pixels size; ii) systematic selection of 10 tiles per each study site to create the
reference data; iii) dataset random split in training, validation and testing followed by a split in training and validation dataset
based on sites geographical distribution for a cross-validation; iv) Hyper-parameters definition and training; v) performances
evaluation. Each of the steps is furtherly explained in the following chapters. We selected tiles of 512 x 512 pixels (equivalent
to 25.6 x 25.6 m at 5 cm spatial resolution) as the model achieved better performance, i.e., higher detection rate and accuracy
on all sites, compared to tiles of 128 x 128 and 256 x 256 pixels.



### 2.3.1 Training, validation, and test data

We here used only 3% of the total amount of tiles for training, with the purpose of testing the limits of using a low number of training images on a pre-trained DL model. To build a strong reference dataset we fine-tuned the model using a Meta AI Segment Anything for the creation of individual tree crowns samples (https://github.com/facebookresearch/segment-anything). For the creation of the ground truth all the trees were segmented and labelled by visual interpretation of RGB images, resulting in non-overlapping binary masks. To minimise operator biases photo interpretation was conducted by a single operator. The semi-automatically delineated tree crowns were used to evaluate the model performances in delineation (see Section 2.3.4). At the end of the process, we obtained a dataset with a total of 1,016 individual canopies of different coniferous species (larch trees n = 885, pine trees n = 131). All the segmented crowns were classified and labelled as "trees" regardless of the species due to the similar spectral information.

To generate the training, validation and test datasets, the reference dataset of 100 tiles (512 x 512) was split into 70 % of images for training, 20 % for validation, and 10 % for testing. The split in the three datasets was performed by systematically sampling the 512-pixel tiles in the reference dataset. The tiles were sampled diagonally in order to cover a larger surface of the study area and to minimise spatial autocorrelation. Finally, we assessed the performance of the model using the test dataset, consisting of tiles with which the model was not familiar.

To validate the model transferability, we corroborated the results with a k-fold spatial cross-validation. The dataset was split in nine folds. In each fold nine sites' images were used for both training and validation, while the images of one site were used for testing only. The process was repeated 10 times so that each site was used as the test dataset once. The outputs of the 10 iterations through the entire dataset were finally averaged to achieve a mean F1 score, precision, recall, and average precision value.

### 2.3.2 Model development and hyper-parameter configuration

During training we utilised the Adam optimizer with a learning rate of 0.00025; 128 ROIs per image; 1500 epochs; batch size of 30. We used the R101-FPN configuration as it is reported to have the best speed in training maintaining a high accuracy in instance segmentation (https://github.com/facebookresearch/detectron2/blob/main/MODEL_ZOO.md). To prevent overfitting we kept track of the validation loss in the F1-score every 100 iterations and implemented an early stopping when the F1-score degraded for more than 5 evaluations. The model was trained with data augmentation consisting in random resizing and rotation of the input images.

We predicted tree crowns contours using the tiling process developed by Ball et al. (2023). This method allowed us to create a buffer around each tile, thus avoiding crowns at the edges of the tile to be split. The overlapping crowns resulting from this operation were then filtered by removing those with the lowest confidence value assigned during the prediction. Classified maps were post-processed to remove noise and correct obvious classification errors. The crowns remaining after this cleaning





process were deemed correctly predicted trees. The evaluation of the model performances was computed before the cleaning
process for all the evaluation metrics apart from DET% and IoU (see Section 2.3.4 for details)

**2.3.3 Model performance assessment**

To assess the performances of the DL model we chose four evaluation metrics according to similar studies focusing on
individual tree detection (Beloiu et al., 2023; Dersch et al., 2023; Dietenberger et al., 2023; Xie et al., 2024). The chosen
evaluation metrics were: precision (1), recall (2), F1 score (3), and average precision (4). The F1 score, which measures the
test accuracy, was computed as the weighted average of precision and recall. The closer the F1 value is to 1, the higher the
accuracy of the class prediction. The average precision is computed as the area under the precision-recall curve. It evaluates
the quality of the classifier in retrieving the relevant instances.
To validate the model transferability, we corroborated the results with a spatial cross-validation. (1), (2), (3), and (4) were
computed after each cross-validation and the 10 outputs were averaged to achieve a mean precision, recall, F1 score and
average precision value to be compared with the not-cross-validated results.
Tree maps were evaluated in terms of detection rate (DET%) and delineation accuracy (IoU). DET% is the ratio between the
predicted and the number of trees measured in the field (5). It is computed to evaluate how many objects were correctly
classified out of all the ground truth data. For the evaluation of the DET% we used as reference data all the field-sampled trees
with the exclusion of the ones falling within the training and validation data. The IoU is measured as the ratio between the area
of overlap and the area of union of the ground truth crown and predicted crown (6), it is used to evaluate the precision with
which the predicted crowns were segmented. For IoU evaluation we used as reference data the semi-automatically delineated
tree crowns in the test dataset.

$$Precision = \frac{TP}{TP+FP} = \frac{correctly\ predicted\ trees}{all\ trees\ predictions}, \tag{1}$$

$$Recall = \frac{TP}{TP+FN} = \frac{correctly\ predicted\ trees}{all\ ground-truthed\ tree\ predictions}, \tag{2}$$

where TP are the true positives instances; FP are the false positive instances; FN are the false negatives (number of ground
truth trees that the model did not detect).

$$F1\ score = \frac{precision * recall}{\frac{precision + recall}{2}}, \tag{3}$$

$$AP = n\sum (R_n - R_{n-1}) \cdot Pn\ \text{AP=n}\sum(\text{Rn}-\text{Rn}-1)\cdot\text{Pn}, \tag{4}$$

where $n$ is the number of thresholds; $Rn$ is the recall at the n-th threshold; $Pn$ is the precision at the $n$-th threshold.





$$DET\% = \frac{number\ of\ predicted\ trees}{actual\ number\ of\ trees},\qquad(5)$$

$$IoU = \frac{area\ of\ overlap}{area\ of\ union},\qquad(6)$$

### 2.3.4 Tree attributes assessment

Tree position estimation accuracy was evaluated through a comparison between the field-collected coordinates with ones of
the centroids of predicted crowns. For the evaluation of height estimation, we compared the value of the CHM at the predicted
centroid with the height measured in the field. The evaluation metrics chosen for evaluating the accuracy in tree height and
position were: root mean square error (RMSE) and mean absolute error (MAE). Both these metrics were calculated in
centimetres. RMSE is a standard deviation of prediction errors or residuals (7). The MAE shows how close the ground truth
values and predicted values are to each other (8). It is obtained as the absolute difference between the real value and the
predicted value, hence, it gives the same importance to small and big differences between estimated and predicted value.
Position estimation accuracy was also evaluated in terms of Euclidean distance between the centroid of the predicted crown
and the one of the ground truth crown (9). For tree height estimation accuracy, we also computed the deviation between real
and predicted values calculated both in absolute and relative terms. RMSE, MAE, Euclidean distance and tree height accuracy
were computed only for correctly predicted trees (n = 343) with the exclusion of the trees that fell within tiles used for training
and validation of the neural network (n = 157).

$$RMSE = \sqrt{\frac{\sum_{i=1}^{n}(x_p - x_r)^2}{n}},\qquad(7)$$

$$MAE = \frac{\sum_{i=1}^{n}|x_p - x_r|}{n},\qquad(8)$$

$$Euclidean\ distance = \sqrt{(X_p - X_r)^2 + (Y_p - Y_r)^2},\qquad(9)$$

where $n$ is the number of observations; $x_p, y_p$ are the predicted values; $x_r, y_r$ are the actual values.
We tested tree height influence on the results' accuracy by grouping trees into 3 categories according to their height: small
trees (height ≤ 130 cm), medium trees (130 cm < height ≤ 200 cm) and tall trees (height > 200 cm). Statistical differences
between the three different size classes results were evaluated using a Wilcoxon test with pairwise comparison between the
groups. To investigate how the inclusion in the analysis of trees smaller than 50 cm impacted on the position and height
estimation accuracies, we created a separate size class excluding them (height > 50 cm).



## 2.4 Spatial pattern analysis

Tree maps and extracted tree heights were finally used to investigate tree spatial patterns. We assessed tree distribution patterns by applying a univariate Point Pattern Analysis (PPA) computed through the grid-based software Programita(2014) (Wiegand and A. Moloney, 2004). We used a pair-correlation function $g(r)$, a second-order statistic that is non-cumulative and uses only points separated by a distance r, thus allowing the identification of spatial scales where there are significant interactions among points. We analysed patterns across a distance ranging from 0 to 100 m, that is one-third of the width of the study sites (Rosenberg, 2015). The observed univariate patterns were compared with simulation patterns and confidence envelopes generated by a Heterogeneous Poisson (HP) null model. This null model distributes the points (tree centroids from the tree maps) on the study area with a probability proportional to the intensity map but relaxes the assumption of complete spatial randomness and allows the intensity of the point pattern to vary across the study area. For the generation of the intensity function to be employed in the HP null model we employed an Epanecnikov kernel with enabled edge correction and we set the ring width of the moving window to 5, and allowed only one point per cell.

To test significant departure from the null model, for each analysis we performed 99 Monte Carlo simulations which generated 99% confidence limits (Carrer et al., 2018; Getzin et al., 2006; Petritan et al., 2015). The spatial pattern was defined as randomised, clustered or regular if the $g(r)$ values were respectively equal, greater or lower than the confidence envelopes calculated using Monte Carlo simulations at specific spatial scales. To verify the robustness and significance of the departure, and to avoid incurring in Type I error (if the value of $g(r)$ is close to a simulation envelope the null model may be rejected even if it is true) we used the Goodness-of-Fit (GoF) over the given distance interval (Loosmore and Ford, 2006).

Additional univariate PPAs were also performed for each tree size category in order to gain insights on tree spatial distribution within each dimension class.

To assess the relationship existing between tall and small trees we applied a bivariate point pattern analysis (Wiegand e A. Moloney 2004). We extended the pair-correlation function used before for a bivariate analysis ($g12(r)$), thus allowing us to detect the interactions between the two different classes of trees. The interaction was defined as independent, attraction or repulsion if the $g12(r)$ values were respectively equal, greater or lower than the confidence envelopes at specific spatial scales. For the bivariate analysis we used the antecedent condition null model, with points of pattern 1 (tall trees) fixed, and points of pattern 2 (small trees) distributed in accordance with a HP null model, where small trees are randomly distributed in the neighbourhood of the tall trees.

To investigate potential dynamics of attraction/repulsion among individuals of different sizes we performed the analysis by using the same classes (tall, medium and small trees) previously created. The middle class was used as a dividing element between tall and low trees in order to avoid overlapping groups, and was hence not used in the analysis.

One of the assumptions of the PPA is that objects (trees) are considered as points. However, we decided to test whether the point approximation (canopies centroids) was somehow hindering the spatial relationships between trees. To investigate this



aspect all the above mentioned analyses were performed again using as input data the crowns' shapes taken from the generated
tree maps, hence using the setting for objects of finite size and real shape (Wiegand et al., 2006).
Univariate and bivariate analyses on points approximation and on objects of finite size and real shape were performed for each
site using the same settings and were ultimately combined with the "combine replicates" protocol.

## 3 Results

### 3.1 Tree detection rate, delineation performances and transferability of the protocol

Throughout the evaluation process, the DL model achieved a F1 score of 0.76, a precision of 0.92, a recall of 0.79, and an AP
of 0.68. As corroborated through the spatial cross-validation, the DL model showed good performances on yet-unseen data,
achieving an F1 score 0.68, a precision of 0.90, a recall of e of 0.56 and an AP of 0.36 (appendix B).
As shown by DET% results, the DL model was able to accurately detect 67% of all the trees sampled in the field (with the
exclusion of the trees present in the training and validation dataset) (Table 2). Small trees were harder to detect, a result that
was reflected by the mean detection rate of 52%. As emphasised by DET% ab50, limiting the analysis to taller trees lead to
higher detection rates, resulting in a DET% = 70, thus confirming our hypothesis that smaller trees have a strong negative
effect on the detection rate.  Considering only ab50, Genevris was the site in which the best detection rates were registered
(93%), followed by Valfurva, Devero, Bocche and Livigno, where the model correctly detected more than 78% of all the trees.
Considering only tall trees we reached a mean detection rate of 86%, furthering supporting the effect of size on detection rates.
According to the IoU results, tall trees were the ones achieving the best performances, reaching a mean IoU value of 0.85.
Medium and small trees achieved a mean IoU value of 0.73 and 0.69, respectively. The difference between tall trees' IoU and
the other two classes' one was significantly different, as demonstrated through a Wilcoxon test (Fig. 4a).





**Table 2. Single site detection rates and number of total predicted trees (n. pred trees) out of the totality of trees sampled in the field (n. test trees). DET% all = detection rate on the totality of individuals; DET% small = detection rate on small trees; DET% medium = detection rate on medium trees; DET% tall= detection rate on tall trees; DET% ab50 = detection rate on individuals taller than 50 cm.**

| site | n. test trees | n. pred trees | DET% | | | | |
|---|---|---|---|---|---|---|---|
| | | | all | small | medium | tall | ab50 |
| Avic | 42 | 14 | 33 | 12 | 56 | 75 | 37 |
| Becco | 45 | 31 | 69 | 58 | 69 | 85 | 71 |
| Bocche | 50 | 35 | 70 | 48 | 85 | 93 | 79 |
| Chianale | 51 | 32 | 63 | 43 | 73 | 68 | 63 |
| Devero | 40 | 33 | 83 | 71 | 86 | 94 | 83 |
| Genevris | 40 | 37 | 93 | 86 | 1.00 | 92 | 93 |
| Livigno | 50 | 39 | 78 | 85 | 63 | 89 | 78 |
| Rion | 45 | 24 | 53 | 18 | 78 | 93 | 57 |
| Senales | 47 | 24 | 51 | 16 | 40 | 83 | 58 |
| Valfurva | 49 | 40 | 82 | 84 | 76 | 86 | 82 |
| **Mean** | / | / | **67** | **52** | **73** | **86** | **70** |

## 3.2 Tree attributes estimation

The proposed method demonstrated that it was possible to accurately estimate tree positions and height. Trees' predicted position achieved a RMSE of 0.59 m and MAE of 0.49 m. For most of the predictions, the calculated Euclidean distance value was below one metre, with the majority of predictions recording a distance close to 30 cm (Fig. 3b). Position precision increased with the reduction of tree height, thus resulting in the two smaller classes (medium and small trees) having the lowest deviation values between predicted and reference centroids (Fig. 4b). Small and medium trees position prediction accuracy achieved a mean Euclidean distance value of 0.40 and 0.44 m, respectively. The Wilcoxon test highlighted a significant difference between the two smaller classes' results and the one obtained for tall trees, for which the mean Euclidean distance value was 0.61 m.

In regard to height estimations, despite some larger errors (outliers) between field and extracted values, a strong ($R2 = 0.87$) linear relationship between the two sets was observed (Fig. 3c). The coefficient of determination, the RMSE of 91.60 cm, and



the MAE of 71.76 cm confirm that the SfM-derived point cloud can be used to accurately estimate tree heights. Almost all
height predictions showed a relative deviation from ground truth values below one metre, and the highest frequency of
predictions recorded a relative deviation close to 0.20 m (Fig. 3d). Tall trees' height was better estimated than smaller ones'
(Fig. 4c). The mean deviation between predicted and real height showed its minimum for the tall class with a value of 0.23 m,
followed by a 0.47 m value for the medium class, and had its lowest accuracy for the small class with a value of 0.62 m.



**Figure 3. (a) Instance segmentation output with a comparison of crowns predicted by the model (shaded with orange outline) and**
**manually delineated ground truth crowns (shaded with blue outline) in Genevris study site. The image illustrates how smaller trees**
**were harder to detect by the model, with some missing segmentations. Kernel density distribution of (b) relative deviation for position**
**estimation and (d) deviation for height estimations. (c) Linear regression model between the field-measured crown heights and**
**estimated heights in metres. The red dashed line represents the 1:1 line.**



**Figure 4. Comparison of model performances in predicting trees (a) canopy surface and shape (measured through IoU), (b) position and (c) height in the three different size classes. Violin plots show the median (black line) and mean (dark red diamonds) values. Statistical differences were calculated through a pairwise Wicoxon test (*: significant difference; **: very significant difference; ***: highly significant difference; NS.: not significant difference)**

### 3.3 Treeline spatial patterns and tree-tree interactions

The univariate analysis resulting from the "combine replicates" protocol (tree crown centroids) showed a strong aggregation over all the sites (Fig. 5a). For spatial scales <20 m there was a marked positive departure from the pair-correlation function indicating clusterization, which turned into a random pattern at 21 m under the HP null model (GoF: $p < 0.05$ in all sites). For spatial scales >35 m the combined replicates showed a slight negative departure from the null model, indicating a regular



distribution. All the sites, if considered separately, showed similar patterns to what resulted from their combination (see details
in appendix C). The univariate analysis conducted on objects of finite size and real shape (tree crown polygons) generated
slightly different results (Fig. 5b). Despite the results still pointing towards a clumped pattern throughout the entirety of the
sites, it appears that the clusterization occurred for all spatial scales from 0 to 100 m. To understand if trees were clusterized
also among the different size classes and not only when considered all simultaneously, we also performed a univariate PPA
for all the tree size classes separately. The results highlighted a clear trend in forming groups at small spatial scales, among
trees of the same size classes (Appendix E).
The "combine replicates" protocol performed on the bivariate analyses (tree crown centroids), through the antecedent condition
null model, evidenced a strong repulsion for small trees in respect to tall trees along all the spatial scales (Fig. 5c). Again, by
analysing each site separately, they all showed similarities among each other and with the combined replicate result (see details
in Appendix D). The bivariate analysis conducted on objects of finite size and real shape (tree crown polygons) led to similar
results (Fig. 5d), suggesting the existence of a strong repulsion between small and tall trees.





**Figure 5.** Univariate pair-correlation function g11(r) for (a) centroids and (b) crown polygons. The analysis allows for the definition of a spatial pattern as clumped, random or regular (hyperdispersed) if the summary statistics (black continue line) value is greater than, within, or lower than the confidence envelope (light grey area). The confidence envelope lines represent the upper and lower 95% simulation envelopes. Black dashed lines indicate the expected pattern if the points showed a random spatial distribution. Correlation analysis of tall trees and small trees for (c) centroids and (d) crown polygons. Values of the g12(r) function that significantly deviates from the null model indicate a significant attraction (if positive) or repulsion (if negative) between the two patterns.





**Figure 6.** Univariate and bivariate PPA results for all study sites along with the fine-scale mapped tree crowns.

## 4 Discussion

### 4.1 Detection performances

We demonstrated that RGB imagery obtained from low-cost UAVs can be effectively used for accurate tree detection across large, heterogeneous areas at elevational treelines. Previous studies have conducted similar analyses employing different segmentation strategies in various forest types. Our model achieved precision and recall values that surpass those reported in other studies (Beloiu et al., 2023; Dietenberger et al., 2023). The average IoU across different size classes was 0.76, lower than results from plantation-based studies (Hao et al., 2021), but superior to those from mixed temperate forests (Dietenberger et





al., 2023). Regarding detection rates and F1 scores, our results fell within the average range recorded in comparable research
(Table 3).
However, direct comparisons with other studies are challenging due to the substantial variability in forest types and algorithms
used. While our analysis outperformed others on certain metrics, it is important to note that our study was conducted in an
environment where detection is facilitated by the reduced presence of intertwined canopies, unlike in tropical or temperate
forests. Conversely, this advantage was offset by the inclusion of small trees in our analysis, a factor that negatively impacted
the results and is often excluded in similar studies.



**Table 3. Recent studies output on tree detection and crown delineation in forest ecosystems using UAV-derived data. DET% = detection rate on the totality of individuals; IoU = Intersection over Union; AP = Average Precision.**

| reference | Forest type | sensor | crown detection algorithm | DET% | precision | recall | F1-score | IoU | AP |
|---|---|---|---|---|---|---|---|---|---|
| Present Work | mixed open woodland | RGB | Faster R-CNN | 70 | 0.92 | 0.79 | 0.76 | 0.76 | 0.68 |
| Beloiu et al. (2023) | mixed temperate forest | RGB | Faster R-CNN | - | 0.75 | 0.78 | 0.76 | - | - |
| Dietenberger et al. (2023 | mixed temperate forest | RGB | Region growing | - | 0.68 | 0.61 | 0.64 | 0.44 | - |
| Weinstein et al. (2019) | mixed open woodland | RGB, LiDAR | RetinaNet | 82 | - | - | - | - | - |
| Xiang et al. (2024) | several forest types | LiDAR | 3D CNN | - | - | - | 0.85 | - | - |
| Dersch et al. (2023 | coniferous, deciduous, mixed stands | LiDAR | Mask R-CNN | - | - | - | 0.86 | - | - |
| Jing et al. (2012) | mixed forests | LiDAR | Multi-scale analysis , Marker-controlled watershed segmentation | 69 | - | - | - | - | - |
| Ball et al. (2023) | tropical forests | LiDAR | Mask R-CNN | - | - | - | 0.64-0.74 | - | - |
| Xie et al. (2024) | Chinese fir plantation | RGB | Mask R-CNN | - | - | - | - | - | 0.55 |
| Hao et al. (2021) | Chinese fir plantation | RGB | mask R-CNN | - | - | - | 0.85 | 0.91 | - |

Our hypothesis was that tree height may have a strong impact on the model performance. Dividing trees with an height criteria allowed us to track detection performance, showing that accuracy improves with tree size across all study sites. In all the study sites, detection was high for taller trees (86%) but low for smaller ones (52%), confirming our hypothesis. In addition to being inherently more challenging to detect in the imagery due to their diminished size, smaller trees often present altered lighting conditions due to being partially obscured or completely concealed by taller ones (Beloiu et al., 2023; Dietenberger et al., 2023; Hamraz et al., 2017), leading to missed detections (i.e., false negatives). This issue is exacerbated in dense clusters (Vauhkonen et al., 2012), highly present in most of our study sites. Another critical challenge in tree detection is the blending





of canopies colours with the background, a factor that largely depends on the tree, shrub and, herbaceous species on the site
(Diez et al., 2021; Weinstein et al., 2019). Here, although the problem also affects tall trees, it was markedly more problematic
for smaller ones.
Despite the most recent advancements in AI tools for object detection and segmentation, the accurate detection of small trees
in RGB images over large areas is still in its infancy and remains unfeasible without a drastic change in flight parameters (very
low flight heights) resulting in significantly extended survey times (especially in mountainous areas where topography can
represent a limit) (Fromm et al., 2019). Nevertheless, due to the harsh environmental conditions at the treeline ecotone, long-
term survival of small trees is jeopardised by factors such as unsuitable sites for survival (Davis and Gedalof, 2018; Marquis
et al., 2021), failure to grow in harsh conditions (Crofts and Brown, 2020; Frei et al., 2018; Müller et al., 2016) and predation
(Brown and Vellend, 2014; Cairns et al., 2007). Therefore, precise mapping of small trees can be considered of secondary
importance if compared to taller and potentially permanent trees.
With the present work we investigated how unique treeline characteristics influenced the model's performance. On Mont Avic
treeline, where European larch is the dominant species, the leaf-off effect on detection rate was tested and the scarcity of green
needles on the canopies resulted in evident worse performances (Table 2). This finding is consistent with previous works
underscoring how leaf-off season surveys are often correlated with lower detection accuracies (Imangholiloo et al., 2019). In
Rion, sunlight condition's effects on the performances of the approach were tested. As already documented in other studies
(Diez et al., 2021), the presence of large and elongated shadows led to a notable decline in detection rates, probably due to the
substantial difference in the colours of the tree crowns in this site's images. These results highlight limits of RGB-based
approaches that still have to be tackled, underscoring the need of applying a standardised sampling protocol throughout all the
study sites to augment results reliability or provide more input data to increase variability in the training dataset.
Rion and Avic excluded, a clear waning trend in tree detection related to a specific terrain feature of the site - presence of rocks
(Becco), herbaceous species (Chianale) or others - was not found. We therefore hypothesise that terrain characteristics had a
negligible - or did not have any- influence on detection rates, supporting the generalizability and transferability of the approach
to treeline displaying different features.

## 4.2 Tree attributes estimation and transferability of the protocol

The proposed approach has proved to be capable of accurate georeferencing and height estimation. Despite the high precision
of the GNSS antenna employed, some small georeferencing errors are inevitable (e.g. due to limited sky view, precision can
be limited). Additionally, during field data collection, GNSS points coordinates (tree's location) are hardly collected directly
below the real treetop, but rather near the corresponding tree locations, thus creating a second inevitable error that adds up to
the previous one (Shimizu et al., 2022; Vauhkonen et al., 2012). Our tree position estimations were highly satisfying and
comparable with results obtained in other recent studies employing similar or more sophisticated gears in environments with
analogous open stands. Castilla et al. (2020) georeferenced coniferous species in a boreal forest using SfM point clouds





achieving an RMSE of 20 cm. Another consistent result is the one of Fernández-Guisuraga et al. (2018) who extracted tree
position of coniferous species in a post-fire environment attaining a RMSE < 30 cm.
Tree height estimations presented a trend skewed towards underestimation, an issue attributable to the low sharpness of the
DSM generated through SfM, as also evidenced by Panagiotidis et al. (2017) and Wallace et al. (2016). Airborne laser scanning
is the most well-known tool for DTM modelling due to its better capability in penetrating tree crowns, which often result in
highly accurate estimation of tree features. However, in the present study we provide evidence that by means of
photogrammetric point clouds it is possible to extract tree height with an accuracy as high as the one obtained through the
employment of LiDAR sensors, which are still moderately expensive, thus limiting the feasibility of repeated surveys in many
cases. Coops et al. (2013) assessed tree height over a Swiss treeline ecotone by employing LiDAR sensors with an RMSE= of
0.70 m. Studies employing LiDAR technologies in boreal treelines documented a standard deviation of 0.11–0.73 m Næsset
and Nelson (2007) and of 0.16–0.57 m Næsset (2009). A study that clearly surpasses our result is that of Wallace (2012), who
determined a mean height standard deviation of 0.24 m in a stand with sparse trees. In contrast, our results are more accurate
when compared to studies employing SfM point clouds to determine tree height. Wallace et al. (2016) investigated the
differences between LiDAR and SfM point clouds in a stand characterised by spatially varying tree canopy cover. The former
achieved an RMSE of 0.92 m, while the latter yielded an RMSE of 1.30 m, a slightly poorer result than ours. Brieger et al.
(2019) estimated tree height in an open larch forest and reported a mean RMSE of 1.42 m, again confirming the higher accuracy
shown by our results when estimating tree height in open stands through photogrammetry.

**4.3 Spatial pattern and trees' interactions on the Italian alpine treeline**

To the best of our knowledge, there are no previous studies that have simultaneously investigated the patterns of multiple
treelines throughout the Alpine range. Several recent studies have highlighted how tree spatial patterns vary along an
elevational gradient within the treeline ecotone (Garbarino et al., 2020; Jia et al., 2022; Wang et al., 2021), however, a study
investigating such patterns over large extents across multiple sites simultaneously is unprecedented.
We found a discrepancy between the univariate analysis performed on centroids (point approximation) and tree crowns
(polygons). The dissimilarities are potentially due to a systematic effect in the size of the objects (Wiegand personal
communication). First of all, the polygon pattern analysis uses more data points (each cell belonging to an object is counted as
a point), and therefore it is possible that the range of significant effects is larger. Furthermore, it is possible that having larger
objects in a region of the observation window, as it is common in our study areas, may result in a greater clumping across the
analysed spatial scale. Such differences in polygon and point summary functions have already been found in previous studies
and are believed to be due to ecological processes (i.e. competition) instead of systematic effects (Vacchiano et al., 2011).
Whether the cause is one or another has to be further investigated.



Despite the discrepancy on the spatial scale, univariate PPA results revealed a tendency towards a clustered horizontal structure
among all trees within our study areas. This is the typical behaviour within the sub-alpine altitudinal belt, as also consistently
found in other studies conducted on elevational treelines in Europe (Beloiu and Beierkuhnlein, 2019), USA (Garbarino et al.,
2020) and China (Jia et al., 2022). Human impact has been the major driving force in shaping the investigated treelines,
affecting patterns and reciprocal patterns of mature and young individuals. However, over the last few decades, the
abandonment of remote areas has led to a decrease in human activities such as grazing and silviculture (Anselmetto et al.,
2024). As a consequence, recolonization processes driven by natural dynamics have become more important. Therefore, it is
possible that the aggregation patterns found may be a result of a recolonization process (Didier, 2001) and of active niche
selection (Maher et al., 2005). Various researchers emphasise how terrain features such as microtopography and soil spatial
patterns can significantly influence tree distribution at the treeline (Feuillet et al., 2020; Marquis et al., 2021; Müller et al.,
2016). The great heterogeneity of terrain inherent to alpine treelines generates a great diversity of microsites, resulting in a
mosaic of favourable and unfavourable niches (Davis and Gedalof, 2018; Marquis et al., 2021). Owing to this, trees can be
rather diffuse on a favourable area but also clustered in small groups where better chances of survival are found. In addition
to topography, competition and facilitation dynamics between tree species may exert an important role on the evolution of the
treeline ecotone. The results of our bivariate tree-tree interaction analysis showed a repulsion between small – potentially
younger – and tall - potentially older - trees at all the analysed spatial scales. This suggests that within the studied alpine
treelines, favourable sites for germination may have undergone progressive recruitment by groups of seedlings over time,
resulting in the different, evenly sized clusters found (Beloiu and Beierkuhnlein, 2019; Wang et al., 2021). This hypothesis is
also supported by the results of the univariate PPA for the separated size classes, which show how trees belonging to the same
size class are organised in clusters in the landscape. The abrupt spatial segregation between tall and small trees suggests that
the dynamics of tree establishment in the studied areas is significantly driven by competition, with small trees favouring sites
far from existing clusters of tall trees. How biotic interactions may play a dominant role in driving treeline encroachment
dynamics has been discussed in previous studies (Frei et al., 2018; Neuschulz et al., 2018). It is broadly known that in
temperature limited environments tree patches can improve microsite conditions, by influencing snow thickness, soil
properties, microclimate and offering physical support and protection from herbivores  (D'Odorico et al., 2013; Germino et
al., 2002). These positive effects, however, can be offset by competition for vital resources such as light, soil moisture and
nutrients (Frei et al., 2018; Moir et al., 1999), which ultimately hinders seedling growth and survival. Although our bivariate
analysis result suggests the presence of competition between size classes – potentially age classes - in high altitude





environments in the Alps, and is in line with previous studies findings (Carrer et al., 2013), further analyses are needed to
advance our understanding of the effects of biotic interactions on tree spatial pattern at the treeline.

**4.4 Limits and perspectives**

We show that by combining low-cost UAV and sensors with open AI libraries, it is possible to accurately map and extract
single tree attributes at a fine-scale. Our detection rates were comparable or superior to many other DL-based classification
studies in natural forests. Nevertheless, recognising small individuals with high accuracy in RGB images remains a challenge.
As highlighted in several recent studies, LiDAR-informed segmentation strategies could provide a valuable alternative for
comprehensive mapping of individual trees, filling the gap left by our methodology. Another crucial feature that is of great
importance for many ecological analyses is the species composition of the community. The use of multi or hyperspectral
sensors enables the detection of tree species and thus analyses of the species composition of stands, interactions among
individuals and spatial patterns of individual and interacting species.
Although in alpine environments natural dynamics have become predominant as a consequence of land abandonment, the
current treelines pattern and structure are legacies of the past human activities. We therefore stress the importance of studying
these human shaped environments in long-term monitoring research. For this task, we envision future research activities to
apply the presented approach to simultaneously map and detect tree species at the treeline. The final goal is creating a global
network of accurately mapped treeline datasets to monitor the effects of global change on treeline dynamics and explain the
position and pattern of the treeline at different scales.

**5 Conclusions**

We tested the performance of a Mask R-CNN deep learning model in capturing single-tree scale attributes in sprawling, remote,
heterogeneous treeline ecotones on UAV derived structure-from-motion point clouds. UAV employment allowed us to conduct
surveys in a more labour and time efficient manner than pure ground-based ones. Retrieving such data over large areas
enhances the representativeness of the investigated sites and thus the reliability of the results regarding ecological processes at
the treeline. Our results showed that the proposed approach can effectively produce fine-scale tree maps over 90 ha of treeline
ecotones. The model performed well by identifying 70% of trees taller than 50 cm and 86% of trees taller than 2 m across the
10 study sites in the Italian Alps. Beyond its success in detecting tree crowns, the approach also performed well in delineation
tasks. We demonstrated the potential of applying the resulting dataset in ecological applications by analysing spatial patterns
and interactions among trees of different size classes.



The present work underpins the possibility of using UAVs to foster treeline studies and thus move away from reliance on data
collected on the ground. The ability to achieve such results with the low-cost equipment used here makes this approach
accessible to a wide range of scientists and forest managers. The adaptability of the protocol to unique study sites' features
with minimal data preparation procedures further enhances its flexibility and versatility, making the methodology valuable for
numerous applications such as forest assessment, restoration and conservation projects.



**Appendix A:**

**Figure A1. Detail in the UAV-derived orthomosaic of (a) Avic, (b) Becco, (c) Bocche, (d) Chianale, (e) Devero, (f) Genevris, (g) Livigno, (h) Rion, (i) Senales and (j) Valfurva.**





**Appendix B:**

**Table B1. Results of spatial cross-validation analysis.**

| site | F1-score | precision | recall | AP |
|---|---|---|---|---|
| *Avic* | 0.60 | 0.83 | 0.48 | 0.14 |
| *Becco* | 0.81 | 0.80 | 0.87 | 0.45 |
| *Bocche* | 0.48 | 1.00 | 0.35 | 0.34 |
| *Chianale* | 0.73 | 0.85 | 0.40 | 0.36 |
| *Devero* | 0.63 | 0.93 | 0.54 | 0.27 |
| *Genevris* | 0.76 | 0.97 | 0.66 | 0.45 |
| *Livigno* | 0.78 | 0.94 | 0.50 | 0.58 |
| *Rion* | 0.62 | 1.00 | 0.50 | 0.34 |
| *Senales* | 0.60 | 0.88 | 0.49 | 0.41 |
| *Valfurva* | 0.78 | 0.76 | 0.84 | 0.32 |
| ***Mean*** | **0.68** | **0.90** | **0.56** | **0.37** |



**Appendix C:**

**Figure C1. single sites' results of the univariate pair-correlation function g11(r) in (a) Avic, (b) Becco, (c) Bocche, (d) Chianale, (e) Devero, (f) Genevris, (g) Livigno, (h) Rion, (i) Senales and (j) Valfurva using point approximation. The confidence envelope (light grey area) represents the upper and lower 95% simulation envelopes. The found spatial pattern is considered clumped, random or regular (hyperdispersed) if the summary statistics (black continue line) value is greater than, within, or lower than the confidence envelope.**

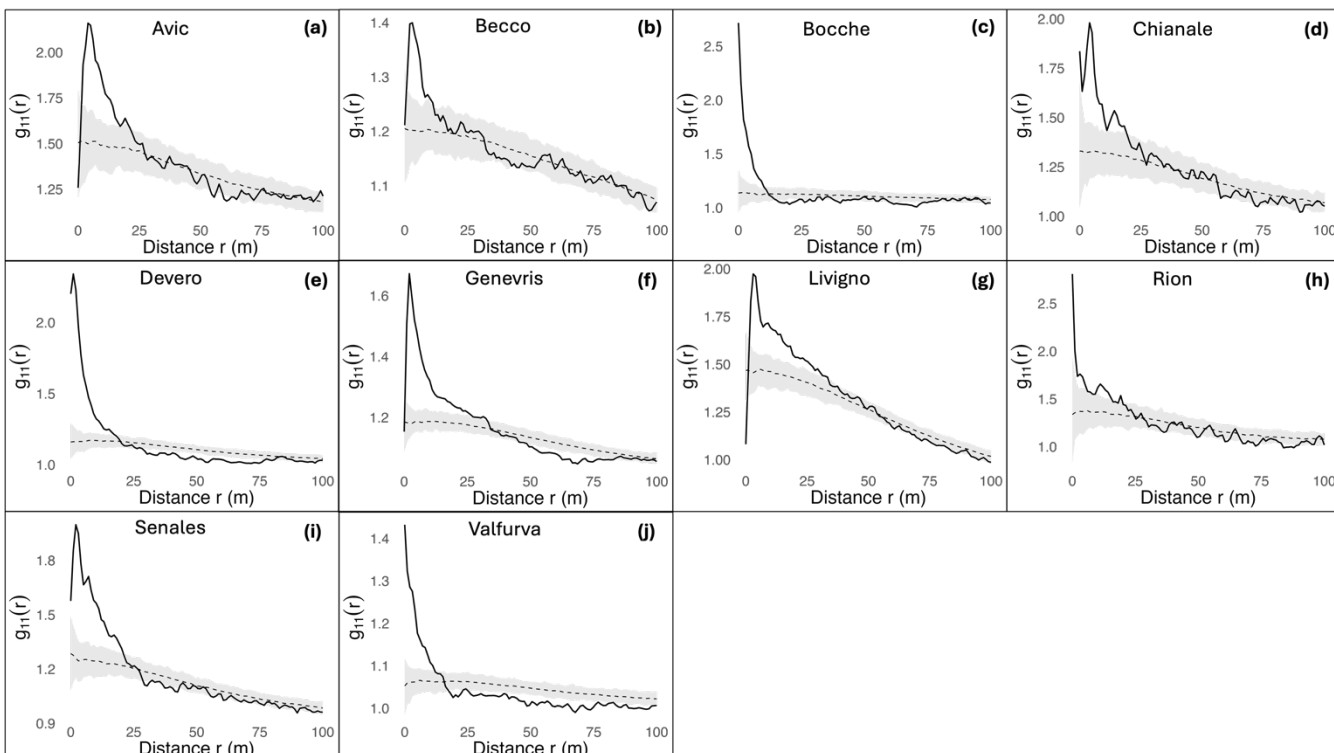





**Appendix D:**

**Figure D1. single sites' results of the bivariate pair-correlation function g12(r) on (a) Avic, (b) Becco, (c) Bocche, (d) Chianale, (e) Devero, (f) Genevris, (g) Livigno, (h) Rion, (i) Senales and (j) Valfurva using point approximation. The confidence envelope (light grey area) represents the upper and lower 95% simulation envelopes. Deviation from the null model (simulation envelope) of the summary statistics (black continue line) indicates a significant attraction (if positive) or repulsion (if negative) between the two patterns**.

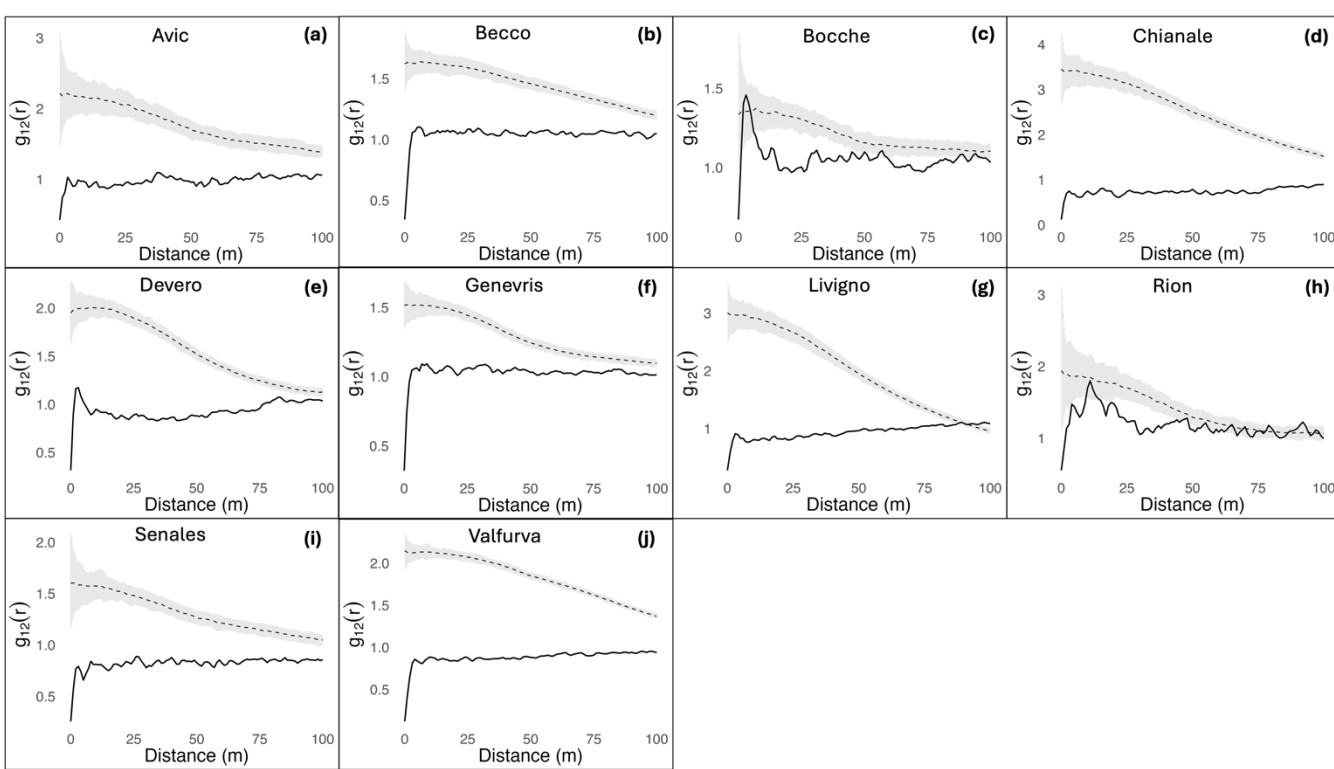





**Appendix E:**
**Figure E1. Univariate pair-correlation function g11(r) for centroids of (a) tall trees, (b) medium trees and (c) small**
**trees. The analysis allows for the definition of a spatial pattern as clumped, random or regular (hyperdispersed) if the**
**summary statistics (black continue line) value is greater than, within, or lower than the confidence envelope (light grey**
**area). The confidence envelope lines represent the upper and lower 95% simulation envelopes. Black dashed lines**
**indicate the expected pattern if the points showed a random spatial distribution.**

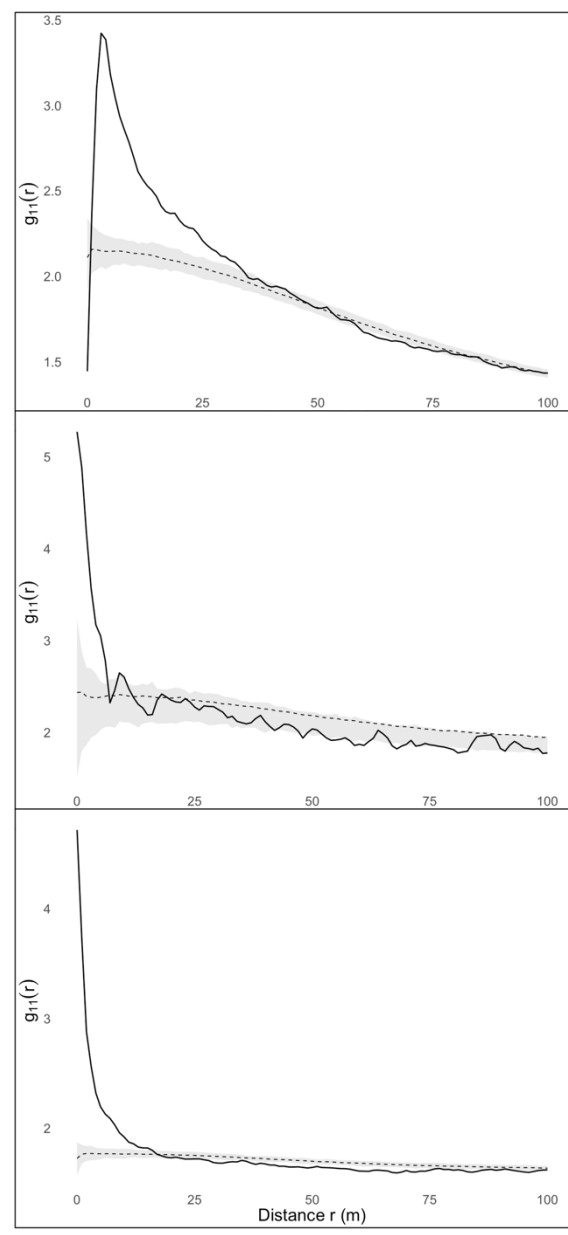




**Code availability**
The code used in the analysis of this research is available upon request from the first author.
**Data availability**
The data used in this research are available upon request from the first author.
**Author contribution**
Carrieri Erik: Methodology, formal analysis, investigation, data curation and writing—original draft preparation. Morresi
Donato: Conceptualization, methodology, formal analysis, investigation, data curation, supervision, writing—review and
editing. Meloni Fabio: Data collection, data curation, writing—review and editing Anselmetto Nicolò: Conceptualization,
methodology, investigation, data curation, supervision, writing—review and editing. Lingua Emanuele: writing—review and
editing. Marzano Raffaella: writing—review and editing. Urbinati Carlo: writing—review and editing. Vitali Alessandro:
writing—review and editing. Garbarino Matteo: Conceptualization, methodology, investigation, funding acquisition,
resources, supervision, writing—review and editing.
**Competing interests**
The author Garbarino Matteo is Editor of the special issue "Treeline ecotones under global change: linking spatial patterns to
ecological processes" to which the paper is submitted.
**Special issue statement**
This article is part of the special issue "Treeline ecotones under global change: linking spatial patterns to ecological processes".
It is not associated with a conference.
**Acknowledgements**
This research was funded by the Ministero dell'Università e della Ricerca through the "OLYMPUS - Spatio-temporal analysis
of Mediterranean treeline patterns: a multiscale approach" PRIN-2022 project #20225S47P8.





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
