# Peer review of "Very-high resolution aerial imagery and deep learning uncover the"

_EGUsphere, 2024_

## Referee Comment (RC2)

[referee-annotated manuscript omitted]

---

## Editor Comment (EC1)

Highlight

Alpine treeline ecotone patterns can indicate how montane forests may, or may not, move up into the alpine zone. This paper presents a low-cost method to create detailed maps of ecotone tree cover, leveraging recent developments in drone-based remote sensing and AI-based image analysis, which has the potential to support ecological studies not only at alpine treeline but in any open woodland or wood-grassland ecotone.

Dear Erik and colleagues,

Thank you for addressing my and the other reviewer's comments on your previous manuscript version. The new version has indeed improved, although some of my points were not solved by the changes made, as detailed below.

Your answer to my comment, that it would be interesting to see what the inaccuracy in tree detection means for the resulting spatial pattern metrics does not really address this point. What you would need to do to address it, is calculate the point patterns with and without the omitted trees and see how much they differ and whether the deviation is acceptable. How to decide whether it is acceptable? I guess that depends on what the differences mean for the ecological interpretation.

The other thing I am wondering about is whether it made sense to combine the point pattern analyses. If your goal is to understand pattern-process relationship, you would want to explore how the sites differ in their spatial patterns (the why would be a separate paper, I agree), not merge them.

Detailed comments:

I am referring to the version in which you marked your changes in green. As in the previous version, yellow markings in the pdf indicate sections that I think need to be adjusted.

L11-13 This sentence makes no sense: which "several" treeline studies, and what is with all other treeline studies? What kind of modelled spatial patterns are you talking about? And what is the studies" and "findings" you might be referring to, since no research question or goal has been presented.  So, similarly, in L14, it is still not clear for what these points are limitations (as it was not before when the word was "gaps"), or what the "implications" in L16 refer to.

L53 Check for consistent use of "elevational" instead of "altitudinal"

L54 The "alternatively" does not fit, because the two sentences it connects both talk about neighbour interactions

L59 It still sounds weird to me to write that heterogeneity in patterns would be a problem for studying pattern-process relationships. It is what you NEED to study such relationships. You correctly write this now, but then what are you referring to when you write "constrains the extrapolation of case-specific observations, thereby limiting their broader ecological generalization."? Maybe you need to give an example to make clear what you are thinking of here.

L61-62 There are a lot of fancy adjectives in this sentence, but it is not clear what the combination of ground-based and rs data has to do with the fundamental question of how to tackle the "spatial heterogeneity issue".

L93-96 The "despite" does not fit here.

L99 These are not really hypotheses that can be formally tested. You could instead call them "assumptions" or "expectations". (in iii the "that" should be removed to achieve a parallel list structure.) L 380: hypotheses are not confirmed (they may be supported etc) and this one was not even really a hypothesis, so you could write something like "confirming our expectation".

L111-112 How can soils include xeric and mesic regions? I would expect to see soil types here, but as far as I am aware, a xeric region is a dry region and does not describe a soil type..

L115 MAT is the least ecologically-relevant temperature information possible. Maybe you can report annual ranges (e.g. of mean monthly temperatures) instead? Or information on snow cover or something else that may affect ecotone pattern.

L132 locations

L142-144 Sorry, but I do not understand this method of mitigating spatial resolution loss. Did you fly the entire plot at two altitudes? And then you used the data from the lower altitude for the lower part of the plots..? And what does the central position in the plot have to do with it? What was in a central position? The drone pilot?

L146 canopy detection

L155-156 I am probably missing something, but don't you need the terrain model to be able to decide whether points are above or below 10 cm from the ground? This procedure seems a bit circular…

L160 With dwarf mountain pine you probably mean Pinus mugo, the krummholz-forming species. It would not hurt to mention these two terms here.

L184 I have a problem with calling these training polygons "ground truth", because they obviously are not based on knowledge from the ground but on an alternative classification method. And you do have actual ground-truth data to work with too. Perhaps you can acknowledge this not quite correct use of the word "ground truth" by putting the term in quotation marks the first time, or by calling them training /validation polygons rather than "ground truth".

L185 annotation of the segments into tree and non-tree classes? Please explain the link between the use of segment-anything and the manual process more clearly.

L198 What do you mean by "from start"?

L250 & Figure 4. I think that here something got misinterpreted from the previous review comments. I think the point is not so much how >50-cm trees affected the position and height estimation accuracies, but also and especially how they, or the accuracy in general, affect (or not) the conclusions based on the point pattern analysis. I.e. taking the validation and evaluation of the model quality one step further.

L287 ""combine replicates" protocol" – I do not know this protocol, but this sounds like you are treating your sites as simple replicates rather than as the units providing the variation you are interested in…

L297 further demonstrating

L324 "Frequency and smoothed kernel density distribution"

Figure 4. I am not sure why this figure was added, but if it was because of one of my comments, that was a misunderstanding (see my comment above).

Figure 5. explain in the caption whether this is for one of the sites, or some sort of average (but see my comment above about the "combine replicates" protocol…

L336 & L346 Similarly, you here claim that the pattern is seen across study sites, but actually you do not, at this point, analyse the sites individually, you just take an average, which I find a strange choice if aiming at understanding pattern-process relationships.

L340 Appendix C is now figure 6?

L354 is this really called a correlation analysis? Also, instead of "of tall trees and small trees" maybe use "between small and large trees"?

L389 Check sentence. What exactly is in its infancy?

L390 Lowering flight height would also increase survey time in other habitats, wouldn't it? Also, in this section there seems to be a confusion between data quality issues and AI abilities.

L419 With "gears", do you mean equipment? Sensors?

L442 This is a very weak argument if you specify your grain and extent so precisely… In the preceding paragraph, make more clear what these studies (not works) did and how it relates to your study. E.g. also using remote sensing, also studying several sites, etc., but not…..

L481 Here, or somewhere else, you should explain clearly what sets your method apart from methods used in the many studies with which you compare you classification accuracy. What makes your method better / different?

L496 lots of adjectives. What do you mean with "sprawling"?

---

## Author Comment (AC1)

**Responses to Reviewer #1: RC1**

Thank you for the valuable assessment and insightful comments to the manuscript. Below, you can find the answers and explanation to all points you raised. All comments were incorporated in the revised manuscript. On behalf of the authors,

Erik Carrieri

**General comments:**

**Reviewer#1:** Your work is important and will likely be very impactful! You demonstrate the effectiveness of using UAV imagery in combination with pre-trained deep learning models for 1) detecting and delineating tree crowns in the ATE, and 2) estimating tree attributes (position and height). Your field dataset is impressive, covering a wide geographic area of the Italian Alps that is representative of heterogeneity at multiple scales and with respect to important climatological, biological, and topoedaphic variables.

**Response:** We thank the reviewer for the positive evaluation and the careful read.

However, your work could be elevated with more nuanced discussion of treeline ecology. You make some general statements in the introduction and conclusion about facilitation and competition, but you do not discuss important nuance related to 1) the degree of stress in the system (especially wind-stress), which can lead to a predominance of facilitative interactions; 2) whether any moisture or nutrient limitations are known to exist in your system that would lead to a predominance of competitive interactions; 3) the species composition of your sites and any important biological factors related to the species present, such as relative tolerance of known stressors at the adult and seedling stages, dispersal modality, and growth rates (or any evidence of age-size relationships in your treeline systems); and 4) anything known about the spatial patterns of variables related to the suitability of sites for colonization, such as distribution of soil characteristics, snowpack, or shelter. You need to discuss the relevant ecological literature in your introduction, use it to inform your hypotheses about the spatial analyses you conduct (which are also missing from the introduction), use it to justify the size classes you delineate in your spatial analysis methods, and finally discuss your results within this ecological context in your discussion.

**Response:** We thank the reviewer for the insightful comment and valuable assessment. In response, we have expanded the Introduction and Discussion sections to include more nuanced discussions of treeline ecology. Additionally, we revised L85–92 to more clearly articulate the study's hypotheses. As Reviewer #2 correctly pointed out, we would like to emphasize that this paper is primarily of a methodological nature, and the point pattern analysis serves as an initial application to demonstrate the approach's potential.

**Reviewer#1:** Secondly, you need to revisit the size classes you use for your point-pattern analyses from the standpoint of data reliability. In your discussion, you make it clear that your model's detection of small trees is biased based on the proximity of those small trees to larger trees. You specifically state that you are more likely to miss the small trees that are closer to large trees. Yet you did a bivariate analysis that relied on accurate detection of small trees to see if they tended to occur close to, or far away from, large trees. Your finding that the small trees tended to be located further from large trees than expected from a relaxed-random distribution could simply reflect the bias in your dataset with respect to the detection of small trees. You absolutely must demonstrate that this is not the case to justify the conclusions you drew based on this analysis. Furthermore, as you discuss the findings of the spatial analyses based on your remote sensing dataset, you must frame these as hypotheses of process based on the observed patterns. There are multiple

possibilities that could explain the observed patterns, with competition and facilitation being among them. I listed many papers that you may find useful for adding this nuance.

**Response**: thanks for the thoughtful comment. In the Discussion section L380-384 we meant to highlight a limit of the deep learning model which we deemed right to emphasize. We agree with the reviewer that the lack of detection of smaller individuals right in proximity of bigger ones might affect the PPA results. However, as it is also visible in Figure 3, the detection of small trees was possible and achieved, even when in proximity of big trees. Attached below are a boxplot and violin plot displaying the distribution of omission errors of small trees in function of the distance to medium and tall trees (distance computed as closest distance between the Ground Truth data and the border of the medium/tall crown). As the graphs show, omission errors are uniformly distributed when related to medium trees. When related to tall trees there is a concentration of missed detection on small scales (average value ~9 m), however, these scales greatly surpass the distance over which a small canopy might be obscured or hidden by a bigger one (and also distance values greater that those we might expect to occur between clonal stems, as mentioned in your next comment below) Concerning the discussion of the observed patterns found, L465-469 & L470-484 already discuss possible reasons of the found patterns but we rephrased and implemented chapters 4.1 and 4.3 for better clarity.

[Figure]

[Figure]

**Missed small crowns distances to medium and tall crowns**

**Reviewer#1:** I want to emphasize that the above does not devalue your study overall, and I very much agree that UAV data fill an important gap in the translation of field data and small-scale processes and patterns at treeline to larger-scale patterns (and potentially processes). Your finding that trees tended to be clustered at scales less than 20 m is very interesting, and seems sound despite the potentially missing small trees. (The opposite result would be less justifiable, given that it could be due to missing trees.) This finding also makes sense based on what is known of processes at treeline (including, but not limited to, facilitation). However, previous spatial analyses of tree patterns in the ATE, done by Elliot et al. (2010), found spatial randomness. Sindewald et al. (2023 – dissertation publication embargoed until Sept. 2025) also found spatial randomness. I would ask that you elaborate on your definition of individual trees. Many conifer species reproduce clonally at treeline, and the stone pine is dispersed by the European nutcracker, which caches seeds. How did you determine which were individual trees and which were clusters of clonal stems? If your cluster analysis was based on discrete "canopies", but those canopies are actually different crowns of a single tree, it would explain why you saw such high levels of clustering at smaller scales as well as why your clusters tended to be of similar sizes.

**Response:** Thank you for your comment. Regarding the potentially missing small trees, we hope the adjustments described in the previous response address your concerns satisfactorily. As for the definition of individual trees, since our study is focused on the detection of tree canopies and builds upon that, we defined individual trees as "*individual tree crowns clearly separable from the other adjacent crowns*" (definition implemented in chapter 2.2 of the revised manuscript). In a purely remote sensing-based analysis like the one employed in this study, it is not possible to determine whether multiple tree crowns belong to the same individual (e.g., clonal stems), which justifies the definition used.

**Reviewer#1:** I would also like to note that Grant Elliot's work comparing spatial patterns across treelines predates your work and compares treelines spanning a greater geographic area (~600 km between the Medicine Bow range and the Sangre de Cristos range). My condolences, but you will

need to walk back your claims in your discussion. Instead, compare how your spatial analysis methods differ from Elliot's and, potentially, why your methods would be better as the standard to use for comparison across treelines.

**Response:** we agree with the reviewer that Elliot's work is indeed an example of a similar study that is comparing treeline on an impressively large geographical extent. However, L421-422 *"a study investigating such patterns over large extents across multiple sites simultaneously is unprecedented."* aimed at emphasizing the overall extent of the study areas (90 hectares), not the geographic range. We aimed at highlighting that our study sites are highly representative of the studied landscapes and no previous analysis was performed with such a level of detail on such large extents. We agree with the reviewer and with reviewer#2 that the sentence can be misleading. We hence rephrased as follows: *Several recent studies have highlighted how tree spatial patterns vary along an elevational gradient within the treeline ecotone (Garbarino et al., 2020; Jia et al., 2022; Wang et al., 2021). Other works have investigated tree recruitment at different sites at broad spatial scales (Nicoud et al., 2025), and others investigated spatial patterns on multiple sites in the Pyrenees (Birre et al., 2023). However, to the best of our knowledge, there are no previous studies that have simultaneously investigated the patterns of multiple treelines at the same level of spatial extent (90 ha) and resolution (5cm) as presented in this work.*

**Reviewer#1:** Regarding your model evaluation, I think you need to add clarity to how you divided your data for training, validation, and testing. It sounded like you were using 70% of the data for training and 20% for validation, then later using those same data within your cross-validation. If this is the case, the results of your cross-validation of the model would not be a true test of model generalizability because the model would have already seen those data during training. Usually, researchers **either** divide their data into training, validation, and testing sets, **or** they do cross-validation and reserve a geographically distinct dataset for testing. I do not understand why you have done both. You can speculate that your model will generalize well based on the variability represented in your dataset, but *if I am correct* that the cross-validation folds included training data the model had previously seen, you cannot use that evaluation to draw conclusions about model generalizability. At the beginning of the section, you also state that you tested the effectiveness of training the model with only 3% of your data (given that the model was pre-trained, off-the-shelf), but unless I am mistaken, you do not report the results of this test. (It would be useful to know how this worked out!)

**Response:** Thank for the thoughtful comment. We modified chapter 2.3: *Our methodology consisted of the following steps: i) cropping the RGB orthomosaic of each study site into adjacent tiles of 512 x 512 pixels; ii) systematically selecting 10 tiles per each study site to create the reference dataset; iii) semi-automatic classification of tree crowns; iv) hyperparameter tuning and model calibration using a dataset randomly split into training, validation, and testing subsets; v) performance evaluation; vi) validation of model transferability through spatial cross-validation.* And chapter 2.3.1 was modified and implemented to clarify the aspect: *To generate the training, validation and test datasets, the reference dataset of 100 tiles (512 x 512) was split into 70 % of images for training, 20 % for validation, and 10 % for testing. […]. The model trained in this way was used to perform predictions on the rest of the tiles to generate tree maps. However, this type of dataset partitioning does not guarantee model transferability since images from all sites are included in each phase of training, validation, and testing. Hence, we performed a spatial cross validation from the beginning to evaluate model generalizability. A k-fold spatial cross-validation was performed using training and validation datasets partitioned according to their geographic distribution. The dataset was partitioned into ten folds based on study sites. In each iteration, images from nine sites were used for training, while the remaining site's images were reserved exclusively for testing. This procedure was repeated across ten iterations, such that each site served as the test set once, thereby ensuring a leave-one-site-out cross-validation scheme.*

We also changed *Figure 2* for better clarity, adding an additional panel displaying the model

transferability testing as a separate phase to the rest of the workflow:

Regarding the use of only 3% of our data for model training, this was not merely a test—this limited subset was indeed the full extent of the training data used, and the reported results reflect the model's performance based solely on that amount.

---

## Author Comment (AC2)

**Responses to Reviewer #2: RC2**

Thank you for the valuable assessment and insightful comments to the manuscript. Below, you can find the answers and explanation to all points you raised. All comments were incorporated in the revised manuscript. On behalf of the authors,

Erik Carrieri

**General comments:**

**Reviewer#2:** You present a useful and successful approach for mapping spatial patterns in treeline ecotones. In this paper, you focus on tree positions and sizes, but I think the general methodology holds further promise for creating extended vegetation maps and terrain models as explanatory variables (a DTM was created as a necessary step to create the vegetation-height model but was not further used at this point for analysing the tree-distribution patterns, it appears).

**Response:** We thank the reviewer for the positive evaluation and constructive suggestion. As noted in the following reviewer's comment, this represents only an initial application of the generated tree maps. Assessing the influence of potential drivers on spatial patterns and processes was beyond the scope of this study, we intend to use these data as explanatory variables in future analyses to better understand the mechanisms underlying the observed patterns.

**Reviewer#2:** The paper is very well written (there are always details, see the yellow markings in the attached pdf and my comments below) and the methodology is generally clear and well presented. The graphs are informative and nicely formatted and the number of figures and tables is adequate. I have just one recommendation that may take some more analyses, and a number of minor recommendations, as listed below, but nothing that cannot be addressed.
Since I took so long to do this review (sorry for that), I can now also refer to the excellent review comments by Laurel Sindewald. I would agree that it needs to be made clearer in the introduction understanding what kind of processes require the mapping of individual trees. However, I understand this paper mainly as a methodological one, and the point pattern analysis as a first application of the results to demonstrate their usefulness. Therefore, I think that the ecological context could be kept quite short and instead the methods could be further validated:

**Response:** thanks for your positive evaluation and the careful read. We are pleased that the reviewer recognized the potential of this work, acknowledging that it is just an initial application. We agree with the reviewer and Reviewer#1 that a clearer introduction in regards of the ecological processes at the investigated treelines is needed. To address this, we have expanded the introduction in the revised manuscript.

**Reviewer#2:** It would be very interesting to see what the inaccuracy in tree detection (of different size classes, as also suggested by Laurel) means for the resulting spatial pattern metrics. This is possibly the most important criterion for deciding whether / for answering what questions the method is good enough: if the resulting point pattern analysis is very sensitive to including or excluding the missed trees (big or small), this would limit the use of the presented method for this analysis, but if it is not, then the method is good enough for such analysis in spite of some inaccuracy. Explaining why clumping / dispersion is an interesting pattern in an ecological sense (e.g. importance of facilitation vs. competition, as also suggested by Laurel) would not hurt in this context.

**Response:** We thank the reviewer for this valuable assessment. In regards of the inaccuracy in tree detection, we report here the same answer that was given to reviewer#2: In the Discussion section L380-384 we meant to highlight a limit of the deep learning model which we deemed right to emphasize. We agree with the reviewer that the lack of detection of smaller individuals right in

proximity of bigger ones might affect the PPA results. However, as it is also visible in Figure 3, the detection of small trees was possible and achieved, even when in proximity of big trees. Attached below are a boxplot and violin plot displaying the distribution of omission errors of small trees in function of the distance to medium and tall trees (distance computed as closest distance between the Ground Truth data and the border of the medium/tall crown). As the graphs show, omission errors are uniformly distributed when related to medium trees. When related to tall trees there is a concentration of missed detection on small scales (average value ~9 m), however, these scales greatly surpass the distance over which a small canopy might be obscured or hidden by a bigger one

[Figure]

[Figure]

**Missed small crowns distances to medium and tall crowns**

**Detailed comments:**

L11-12 make clear what kind studies you are talking about and what "overall pattern" you would want to model

**Response:** thanks, we rephrased the sentence: *However, several treeline studies remain case-specific, focusing on local patterns and processes. Treelines worldwide exhibit a great variability even within single landscapes, which limits the reliability of modeled spatial patterns, and the generalizability of findings based on case studies.*

L14 the "therefore" does not logically lead to conservation…

**Response:** thanks, we have modified the sentences for better clarity. Please see above

L16 what gap?

**Response:** we changed the word "gap" with the word "limitation": *Advancing methods to accurately map fine-scale treeline spatial patterns over large extents is crucial to overcome this limitation.*

L24 It is not clear what is meant with "tend to progressively occupy safe sites" (and the term "safe sites" does not occur anywhere else in the paper…)

**Response:** Our intention was to convey that, over time, trees tend to establish in sites with favorable conditions for growth. This trend was reflected in the PPA, which revealed within-class clustering among small, medium, and large trees. Such clustering suggests that favorable sites were colonized in successive "waves," with large numbers of seedlings establishing simultaneously in the same areas. The sentence was removed to avoid further increasing the already considerable length of the abstract.

L35-36 Citing Körner here (e.g. v would be appropriate

**Response:** thanks, we added the citation.

L42 Mienna et al. did not study carbon sequestration at all. They just mention in the introduction, like many papers do, that treeline dynamics are important for carbon sequestration, but that does not make it a suitable reference for this point.
**Response:** we apologize for the mistake and oversighting. We changed the reference to correct ones.

L43 It is very popular to claim that treelines respond sensitively to climate change, but the interesting thing about them is that some do not respond sensitively at all….
**Response:** We agree with the reviewer that there is an overexertion of the effect that climate exerts on treeline ecotones. The sentence does not imply that all the treelines respond in the same way

L48-49 this sentence is not clear.
**Response:** rephrased to *"However, understanding vegetation changes in response to the complex interplay of these drivers requires studying highly heterogeneous systems across broad spatial and temporal gradients (Holtmeier and Broll, 2007, 2017)."*

L50-51 The great spatial heterogeneity is actually a great chance in the context of understanding pattern-process relationships, because it allows us to relate many different patterns to many potential drivers. If they all looked the same it would be boring. It is not so clear here what you are aiming at with "hinders case-study observations". What would you be trying to generalise here?
**Response:** We absolutely agree with the reviewer. With the mentioned statement we aimed at saying that the high heterogeneity found within the treeline ecotone might hinder the transferability of a process-pattern relationship found in a portion of the landscape to another portion of it (Or even more difficult would be to generalize process-pattern relationship found in a study site to other sites) because of the heterogeneity of the landscape itself. Which is an intrinsic feature of every natural system, including high-altitude ones such as the ones analyzed.
We rephrased and expanded the paragraph:
*In forest ecosystems, tree spatial distributions retain critical signatures of historical dynamics and can be used to derive insights into underlying ecological processes (Grimm et al., 2005; McIntire and Fajardo, 2009; Salazar Villegas et al., 2023). For instance, tree distribution can reveal species-specific coping strategies under stressful conditions, where positive facilitative interactions may prevail (Callaway, 1995, 1998; Smith et al., 2003). Alternatively, tree spatial patterns may encode signatures of intra- and interspecific interactions, encompassing both facilitative and competitive associations (Getzin et al., 2006; Salazar Villegas et al., 2023). Assessing these spatial association patterns among species can help to disentangle the mechanisms shaping treeline structure and dynamics. In this context, the great spatial heterogeneity observed in high-elevation ecotones provides a great opportunity to investigate pattern-process relationships. However, this same heterogeneity constrains the extrapolation of case-specific observations, thereby limiting their broader ecological generalization.*

L51 I recommend "elevation" for height of the terrain above sea level, and "altitude" for height in the air.
**Response:** Thanks for the suggestion. We have modified the word accordingly, here and in the entire manuscript

L58-59 I recommend being a bit friendlier about field data, sv
**Response:** We apologize for the overly rigid wording and have rephrased the sentence to convey a more balanced tone.

L60 The use of remote sensing dates back a lot longer, with aerial imagery available from the start of the 20th century, and ground-based remote sensing even longer!
**Response:** It is true that there is availability of aerial imagery dating back to the 20th century. However, those very pictures found their first application to treeline ecology in the late 1980s.

L63 long time periods rather than wide time intervals (the latter sounds like a low temporal resolution, which is not what you meant)
**Response:** changed

L66 Uncrewed or Unmanned?
**Response:** we aligned the acronym to "uncrewed" in all the manuscript to make it more inclusive

L68 friendly use = user-friendliness
**Response:** thank you, we rephrased: "*Their growing availability and ease of deployment make UAVs increasingly valuable for applications such as detailed tree mapping.*"

L72 Although I really like single-tree scales in treeline ecology, and it is useful and even essential for answering some of the questions, this is not the only possible or useful scale in treeline ecology.
**Response:** With the mentioned statement we did not want to devalue other scales of analysis, we aimed at emphasizing the great importance of tree-scale studies since they provide a level of detail We rephrased the sentence and expanded the paragraph:
*Single-tree mapping approaches are crucial in treeline ecology, as they provide insights into the underlying ecological processes shaping treeline pattern and structure. Seedling establishment - a key driver of plant community dynamics - heavily depends on the presence and availability of microsites that provide suitable conditions for growth and survival (Frei et al., 2018). Multiple local factors such as topography, vegetation, and herbivory influence tree recruitment and thus mediate treeline dynamics (Elliott and Kipfmueller, 2010; Lett and Dorrepaal, 2018; Ramírez et al., 2024). Neighbouring vegetation can either hinder or enhance tree recruitment through competitive or facilitation associations (Getzin et al., 2006; Getzin et al., 2006; Salazar Villegas et al., 2023; Smith et al., 2003). Whether these interactions result in a positive or negative feedback depends on the fine-scale interplay between biotic and abiotic factors. The resulting spatial patterns at the individual tree-scale provide a valuable perspective to both infer past processes and predict future trajectories. Accurate high-resolution single-tree maps are essential tools needed to capture these fine-scale patterns and investigate such tree–tree interactions.*

L80 & L82 These are not real hypotheses. You could write "We aimed to show that…."
**Response:** changed

L85-87 Explain why mapping trees at treeline is different.
**Response:** implemented: "*[...], the distinctive species composition, stratified horizontal and vertical structure, and complex terrain characteristics of treeline ecotones confer a unique ecological identity to these environments.*"

L97-98 Are you sure there is a gradient from oceanic to continental from west to east? In the Alps the climate gradients usually vary from the outer flanks to the inner valleys, and I have never heard that the area of your eastern study sites should be oceanic. Also, the precipitation data in table 1 do not really reflect such a gradient.
**Response:** With L101-109 we aimed at stressing the high variability of features (in climate, topography, morphology, land use practices…) that can be found across the Italian Alps, since our study sites cross the mountain range almost entirely. As it is specified in L109 "*The introduction of*

*such heterogeneity in the dataset was aimed at testing the transferability of the protocol to several treeline conditions".* We modified the paragraph to give a more detailed description of the studied treelines: *Introducing such heterogeneity allowed us to test the transferability of the protocol to several treeline conditions. The selected treelines present elevations ranging between 2100 and 2400 m a.s.l., and variable slope aspects due to the differing orientations of the valleys. Above the closed forest, soils include both mesic and xeric regions and feature patches of grasslands, sparsely vegetated areas, screes, and surfaces shaped by gravitational events such as rill and gullies. All the selected landscapes experienced centuries of human land-use practices under varying intensities of management pressure. In general, land abandonment is more marked in the Western sector of the study area (Bätzing et al., 1996). Across all sites, the mean annual temperature ranges between 0 C° and 2 C°, while the mean annual precipitation varies from 800 mm to 1800 mm.*

L98-99 What are the implications of this northern and southern air; this is not very clear.
**Response:** As above

L99 What do these temperature ranges refer to. The study sites? But that does not fit with table 1. And it would be very warm for a treeline to have an annual temperature of 10°C, if the globally consistent SUMMER temperature at climatic treelines is around 6.4°C…
**Response:** As above

L100 The term "physiography" is unusual and imprecise. What do you mean with it exactly?
**Response:** Removed

L101-102 Here you describe the general regions, but that hardly seems relevant; I would rather expect a description of the treeline sites here
**Response:** changed as above

L102 In what sense a "sequence"?
**Response:** changed as above

L114 shown here are the entire Alps, not just the Italian Alps, right?
**Response:** changed to "the Alps"

L115 consider explaining abbreviations in figure captions to make them more self-explanatory
**Response:** added a description of CHM. We deem UAV to be broadly known as an acronym and no further explanation is needed.

Table 1: I suggest sorting the table by longitude (west to east)
**Response:** Thank for the suggestion, modified accordingly.

Table 1: were the temperatures from CHELSA corrected to correspond with the site elevation?
**Response:** CHELSA dataset already considers topographic conditions to correct for elevation and wind exposition of the valley. The data concerning temperatures and precipitations are provided just for a general description of the sites and the heterogeneity they present. Furtherly, we did not use climatic layers in the analyses, we hence believe a higher level of accuracy is not necessary.

L121 It is a bit hard to believe that your study sites were really randomly selected. Where there no considerations of e.g. accessibility?
**Response:** Changed in "*We selected ten treeline ecotones above 2,000 m a.s.l. along an east-west gradient across the Italian Alps, with a minimum distance of 25 km between sites. Site selection was*

*stratified by administrative region with only fully accessible location included, and edaphic treelines were explicitly avoided."*

L132-133 how did starting from the middle of the plot affect the spatial resolution if the flight height was fixed relative to the highest point of the plot? And why did you choose a fixed flight height rather than following the terrain?
**Response:** We rephrased L133-L137 for clarity. Six flights in two sets of three were performed. The first set was performed at a fixed flight height of 30 from the top of the study area. The second set was performed at a fixed flight height of 30 from the middle of the study site. The 'following terrain' function was not yet implemented in the employed software. Concerning the spatial resolution, as mentioned in L148 the spatial resolution achieved was that of 5 cm. Images of the upper part of the flown areas presented a higher spatial resolution, but they were down sampled to the lowest resolution present in the images. The software automatically down sample the images resolution to the coarser resolution achieved in the images. Modified:
*Each UAV survey consisted of three flight paths: two of them with the camera in the nadiral position (one aligned along the contour lines and the other perpendicular), and one with an oblique camera perspective of 60° off-nadir, granting a more complete view of trees and terrain features. To mitigate spatial resolution loss in the lower portion of the plot due to the slope steepness, each set of three flights was repeated from a central position of the plot, at approximately 150 m from the plot side, resulting in a total of six flights per study site. Flight height was fixed at 30 m above the highest point of the 300 × 300 m plot for the first set and above the middle of the study site for the second.*

L134 "shade patterns from clouds"
**Response:** changed

L135 "respectively, 80 and 80%" -> "both 80%"
**Response:** corrected

L135 "ensure a comprehensive coverage of the surface" -> "to allow calculating robust structure-from-motion outputs" or something like that…
**Response:** changed

L141 What is a "ground classification"? An extraction of the ground (as in terrain) surface from the point cloud? How does this "normalise" the point cloud. For readers not familiar with this procedure, like me, this is not clear.
**Response:** We changed the sentence at L148-151. The metashape software automatically classify each point of the point clouds in "ground" or "not ground" based on a height threshold. All points with a height lower than 10 cm were classified as ground and were used to produce the DTM. Changed: *The classification of ground and non-ground points in the point clouds was based on a threshold of 10 cm height: points lower than 10 cm were considered ground and used to produce the DTM. Canopy height models (CHMs) were then produced by subtracting the DTM from the DSM.*

L142 Some readers may appreciate another definition (full name) of DSM and CHM here
**Response:** clarified

L143-144 you might want to start this sentence with "in the filed work" (since you were talking about processing the point clouds and suddenly jump back to the field
**Response:** clarified

L160 the "reference data" = "dataset" how were these tiles turned into datasets exactly?
**Response:** each tile corresponds to a single image, hence, the complex of all the images is itself a dataset. For consistency, we modified the term "data" to "dataset" where it could be misleading

L161 what do you mean by "sites geographical distribution"?
**Response:** To perform a _spatial_ cross-validation, data were split in training and validation datasets based on their (geographic) spatial distribution. We used all the images from 9 sites for training, and all images from the remaining site were kept outside of the training process and were used to validate the model performance (this process was repeated 10 times, leaving each study site out of the training process once). We modified L167-169 & L187-192 for better clarity: _Our methodology consisted of the following steps: i) cropping the RGB orthomosaic of each study site into adjacent tiles of 512 x 512 pixels; ii) systematically selecting 10 tiles per each study site to create the reference dataset; iii) semi-automatic classification of tree crowns; iv) hyperparameter tuning and model calibration using a dataset randomly split into training, validation, and testing subsets; v) performance evaluation; vi) validation of model transferability through spatial cross-validation._ And: _[...]. The model trained in this way was used to perform predictions on the rest of the tiles to generate tree maps. However, this type of dataset partitioning does not guarantee model transferability since images from all sites are included in each phase of training, validation, and testing. Hence, we performed a spatial cross validation from the beginning to evaluate model generalizability. A k-fold spatial cross-validation was performed using training and validation datasets partitioned according to their geographic distribution. The dataset was partitioned into ten folds based on study sites. In each iteration, images from nine sites were used for training, while the remaining site's images were reserved exclusively for testing. This procedure was repeated across ten iterations, such that each site served as the test set once, thereby ensuring a leave-one-site-out cross-validation scheme._

L167-169 It is not so clear to me how the "reference dataset", created with segment-anything, relates to the ground truth through manual image interpretation. Or how the segment-anything segmentation relates to your trained DL model. Why not just use the segment-anything segmentation directly to predict tree canopies across the study areas, if you assume that that model performs better (since you use it to validate the DL model, L213-214)? In L227 you even mention a "ground-truth crown".... Is this the manual one or the segment-anything one? This logic needs some more explanation.
**Response:** The " strong reference dataset" (composed of canopies segmented with Segment Anything software) mentioned in L167 is the "ground truth" that was created through manual image interpretation. The Segment Anything software takes as input the aerial images and displays them on the monitor. The software **does not** perform any kind of image recognition. The user is fully responsible of "manually" delineating objects (in our case: tree crowns), the software only helps the user to define the borders of the object that is clicked on (this is the reason of hyphens in "manually", because it is not a pure manual delineation process, it is assisted). Hence, the "manually delineated crown" corresponds to the "segment-anything delineated crown" (because the software does not delineate the crowns autonomously) and were used as the "ground-truth crowns" to evaluate the model performance in terms of IoU (L213-214). Said ground truth crowns are the ones visible in Fig3a. For clarity, we have aligned the definition of the "ground truth crowns" in the entire manuscript to avoid possible misunderstanding and we changed the label of "training crowns" to "ground truth crowns" in the "image classification" section.

L174 Did the larches and the pines and spruces really have similar spectral information? That is quite surprising!
**Response:** In terms of RGB channels they do. In particular, discrimination between pines and

spruces was not possible because of the lack of specific traits in the aerial RGB images allowing for a discrimination of the species.

L207 what doe you mean with the "not cross-validated results"? Those where the model was evaluated in the same site where it was trained?
**Response:** Yes. As it is clarified in the revised version, the model performance was tested two times: one time it was trained, validated and tested with a dataset made of images of all the study sites mixed together in all the three folders; another time it was trained and validated using a spatial cross validation logic to test for transferability of the model.

L225 This does not seem correct to me. Sounds to me like the MAE gives more weight to large absolute differences than to small one, but the same weight to the same difference irrespective of the relative differences, i.e. irrespective of the total height of the tree.
**Response:** clarified

L239 Why would a program for point-pattern analysis be grid based? What do you mean by this?
**Response:** thank you for the comment. It was an oversight: the software doesn't rely on a computational grid like grid computing systems, but the data it handles (point locations) can be visualized and analyzed on grids or maps. We did not use aby grid based analysis. We modified accordingly

L246 Can you explain how you let the intensity vary across the study area. I.e. what did the null model look like. I guess the ecotonal gradient from forest to no trees was somehow translated into this null model. But how? Since this is important for understanding the outcome of the PPA, it would be good to explain this in the paper.
**Response:** The intensity vary across the study area accordingly to the chosen null model. The intensity map is a density plot of the observed patter. Under the Heterogeneous Poisson (HP) null model the probability for a point to fall in an area of the plot is greater in areas where there are more trees in the observed pattern. This is different from the null model of Complete Spatial Randomness (CSR) in which the probability for a point to fall in an area of the plot is equal over all the surface of the plot.

L281 I would call this an expectation rather than a hypothesis
**Response:** changed

L298 This Wilcoxon test does not seem to be a very important result. Why did you even want to know whether these accuracies differed significantly? I think you could leave it out and just present the accuracies themselves.
**Response:** We understand the point of view of the reviewer that the mean values reported on the violin plots are enough to provide evidence to the reader of the precision achieved for the different height classes in terms of IoU, height, and position assessment. Yet, the Wilcoxon test makes the analysis more robust in that it provides an estimate of the certainty in extracting a specific metric for a specific size class (i.e. position estimation for the small and medium classes was not as accurate as the tall class one).

L305 Explain what you mean by "better"
**Response:** changed to "more accurately"

Figure 3 Explain what the orange areas are in b and d.
**Response:** explained

Figure 4 Explain abbreviation to make the figure self-explanatory. And if you start explaining what the violin plot shows, you might explain it more completely. How does a relative deviation in m work? Might this be the absolute deviation?
**Response:** figure implemented with a more explanatory caption:

[Figure]

*Figure 4. Comparison of model performance for three tree-height classes (Small: ≤ 130 cm; Medium: >130 cm and <=200 cm; Tall: > 200 cm) in predicting trees (a) canopy surface and shape, measured as Intersection-over-Union (IoU) between predicted and reference crown polygons, (b) position deviation, measured as Euclidean distance (m) between predicted and reference tree centroids and, (c) height relative deviation, measured as absolute difference between predicted and reference height divided by the reference height. Violin plots width at a given value shows the kernel-density estimate of the distribution; the overlaid boxplot displays the interquartile range with the median (black line) and mean (dark-red diamonds). Statistical significance (pairwise Wilcoxon tests) is indicated as: NS = not significant; \* p < 0.05; \*\* p < 0.01; \*\*\* p < 0.001.*

L322 & 327 clusterization & clusterized -> clustering & clustered
**Response:** modified

Figure 6 add information about the spatial scale of the images (i.e. size of the study sites) in the graph or caption.
**Response:** implemented in caption: ***Figure 6****. […] tree crowns overlapped with the 9 ha orthophoto as a background image.*

Table 3 Add: Egli, & Höpke, 2020
**Response:** the mentioned paper, although interesting and outstanding, focuses on a tree species classification task. Hence, we apologize but we deem introducing said paper in table 3 would be

misleading since it does not provide results that can be compared to the listed ones (which focus on tree detection and/or tree crowns delineation)

L378-379 You might also argue that mapping and monitoring the small trees is particularly important to understand the processes going on in the ecotone. This is actually what I expectd as a logical conclusion after the previous sentence…
**Response:** We modified the sentence: *Thus, while the precise mapping of small trees may be of secondary importance compared to taller, potentially permanent trees when evaluating survival rates and seed distribution, small trees are crucial when investigating the encroachment process.*

L381 First time a test of the leaf-off effect is mentioned. This should be introduced earlier.
**Response:** added in chapter 2.2

L384 First time you mention this test too….
**Response:** as above

L386 Do you mean that the detection rate was low in the cross-validation because the image looked quite different from the others due to the low sun angle causing long shadows and different colours?
**Response:** modified: *low sun angles lead to variations in canopy color and the formation of long, distorted shadows, which can significantly impair detection accuracy.*

L390 This is a very promising result. At this point, you would usually "conclude" rather than "hypothesise", although I appreciate the caution with which this statement is made. How about "It therefore appears that …."? This made me wonder: how did you treat Pinus mugo? As a small tree, or as non-tree? (please mention this somewhere). I suspect that sites with lots of Pinus mugo as a matrix among which trees stick out (or not) could cause particular problems. This may be worth mentioning at this point, as an opening for further research into the question of how important background vegetation is.
**Response:** thank you for the positive comment and the insightful suggestion. We modified the sentence as recommended and described how we treated Pinus Mugo (PM) "*Due to its low abundance and specific growth form characteristics (Table 1), dwarf mountain pine was not considered as a tree in our analyses.*". PM presence in our study sites was very limited (see Table 1). As a consequence, it limited classification accuracy only in negligible parts of the study sites and we cannot infer on the effect that it had on tree detection and segmentation.

L394 To make this logic easier to follow, add as second sentence something like: "and some of the deviations may in fact be due to inaccuracy in the ground control data rather than the UAS images."
**Response:** Thank you for the suggestion, implemented: *The proposed approach has demonstrated the ability to accurately georeference individual trees (RMSE = 0.59m; MAE= 0.49m) and estimate their height (RMSE = 91.6 cm; MAE = 71.8 cm); some of the observed deviations may in fact be attributable to inaccuracies in the ground control data rather than the UAV images.*

L403 Since this paragraph presents a lot of numbers, it would help if you would repeat the accuracy in m here. I am not sure that these detailed comparisons of accuracies at the cm scale is very exciting for the readers though. More interesting would be to know why the accuracies differ (whereby actually the differences are not huge and all the same order of magnitude), and why yours would be better or worse than other studies, if this can be explained by your method vs the other SfM methods.
**Response:** We agree with the reviewer this chapter may be tedious. Nevertheless, we believe a

comparison with other results achieved in literature is needed to provide the reader with a general overview of the state-of-the-art in height estimation with UAV platforms adopting different sensors. Unfortunately, an open issue of RS-based studies is the difficulty of cross-study comparisons. Due due to the high quantity of parameters at play (flight height, sensors' specifics, different investigated species, different forest systems, different terrain and background features, etc...) it is arduous to compare different studies. Whereas a study was capable of detecting tree height with a worse/better accuracy than the one we achieved is stated with punctual comments in each sentence in which a study is cited. We repeated the accuracy in m as suggested. We rephrased the paragraph to make it more fluid:

*Using LiDAR, Wallace (2012) reported a mean height standard deviation of 0.24 m in a stand with sparse trees—a level of precision that clearly surpasses our results. However, when compared to studies using SfM point clouds for tree height estimation, our results demonstrate higher accuracy. For instance, Wallace et al. (2016) compared LiDAR and SfM-derived point clouds in a stand with spatially variable canopy cover, finding RMSE values of 0.92 m and 1.30 m, respectively—the latter being higher than ours. Similarly, Brieger et al. (2019) estimated tree heights in an open larch forest and reported a mean RMSE of 1.42 m, further supporting the comparatively greater accuracy of our photogrammetric approach for tree height estimation in open stands.*

L407 Instead of "as high as" a more cautious "not much worse than" may be better, since high-quality laser scanning must reach higher accuracy, and you also give such examples here yourself.
**Response:** Changed.

L422 At this detail and extent and for the Alps it may indeed be unprecedented, but I would indeed recommend toning down a little bit in this section. There are studies in the Alps on e.g. patterns of tree recruitment at different sites (e.g. Nicoud et al 2025), and there are multi-site studies about spatial pattern in e.g. the Pyrenees (e.g. cited papers by Camarero, Batllori, Guttierez et al, or not-yet-cited papers by Ameztegui et al 2016, Birre et al 2023).
**Response:** Thank you for the references recommendations. We agree with the reviewer and with reviewer#1 that the sentence should be rephrased and toned down a bit. Rephrased as follows:
*Several recent studies have highlighted how tree spatial patterns vary along an elevational gradient within the treeline ecotone (Garbarino et al., 2020; Jia et al., 2022; Wang et al., 2021). Other works have investigated tree recruitment at different sites at broad spatial scales (Nicoud et al., 2025), and others investigated spatial patterns on multiple sites in the Pyrenees (Birre et al., 2023). However, to the best of our knowledge, there are no previous studies that have simultaneously investigated the patterns of multiple treelines at the same level of spatial extent (90 ha) and resolution (5cm) as presented in this work.*

L426 The pattern is very different between point and polygon metrics, it is clearly not just an effect of a different range of significance (which, if I understand it correctly, would only result in a wider confidence interval around the null model).
**Response:** We understand the reviewer's confusion in regards of the plot differences. However, we have also contacted Wiegand Thorsten for assistance and he gave an explanation as the one reported in the paper. Essentially, having larger objects in the observation window leads to a greater clustering because of the larger space occupied by each individual. Larger objects (crowns) result in larger clustering if compared to point approximation (centroids) because in some cases the observation window will be almost fully occupied by the objects (crowns).

L429 With "systematic effect" you mean "methodological bias" or something like that?
**Response:** changed

L438 Please explain "active niche selection" for trees (both the "active and the "niche" part)
**Response:** we agree with the reviewer the word "niche" is inappropriate. Changed: *The aggregation patterns found underpin this hypothesis showing that seedling preferentially establish in microsites with better conditions.*

L451 This conclusion may need some more thought. How do you imagine competition may work across 50m? May this "repulsion" relate to a large-scale gradient from established forest at one end of the gradient to recruitment at the other end, or is this taken care of in the null model (please see my previous comment about explaining the handling of the gradient in the null model)?
**Response:** We thank the reviewer for this thoughtful observation. The potential effect of the large-scale gradient is indeed accounted for in the null model used. "*This null model distributes the points (tree centroids from the tree maps) on the study area with a probability proportional to the intensity map but relaxes the assumption of complete spatial randomness and allows the intensity of the point pattern to vary across the study area.*". As explained ina response to a previous comment:
*The intensity vary across the study area accordingly to the chosen null model. The intensity map is a density plot of the observed patter. Under the Heterogeneous Poisson (HP) null model the probability for a point to fall in an area of the plot is greater in areas where there are more trees in the observed pattern. This is different from the null model of Complete Spatial Randomness (CSR) in which the probability for a point to fall in an area of the plot is equal over all the surface of the plot.*
Hence, the model incorporates the underlying spatial heterogeneity, including the gradient from established forest to recruitment zones, ensuring that the observed pattern of repulsion is not an artifact of this gradient but rather reflects a deviation from the expected distribution under such spatial structure.

L467-469 Using good training data, VHR images can also distinguish tree species even if they are RGB or monochromatic, since species have different crown shapes. This may be worth mentioning here too (i.e. the investment can be either in better sensors or in better training data, depending on whether one has more money or more time available).
**Response:** implemented: *Alternatively, species-level analyses are also possible with very-high-resolution RGB images acquired through low-elevation UAV flights achieving a very fine ground sampling density (~ 1.6 cm/px (Egli and Höpke, 2020)), as they can reveal species-specific crown architecture and morphology.*

L487 … on the ground or by expensive aerial surveys or lower-resolution, lower-quality and more expensive satellite imagery.
**Response:** thank you for the comment, changed: *The present work underpins the possibility of using UAVs to advance treeline research, bridging the gap left by limited-in-scale and labor-intensive field surveys and less accurate satellite imagery.*

---

## Referee Report (RR1)

1. We thank the reviewer for the insightful comment and valuable assessment. In response, we have expanded the Introduction and Discussion sections to include more nuanced discussions of treeline ecology. Additionally, we revised L85–92 to more clearly articulate the study's hypotheses. As Reviewer #2 correctly pointed out, we would like to emphasize that this paper is primarily of a methodological nature, and the point pattern analysis serves as an initial application to demonstrate the approach's potential.

**Response:** The additional material in the introduction and discussion is an improvement, though I would not say that it is nuanced given that it is not at all specific to the study area where you were piloting these methods. While I understand that this paper is a methods paper, you chose to try to demonstrate the applicability of the methods with a case study analysis. However, the analysis itself is lacking in direction. It is unclear how/why you chose the size categories you chose, or why they would be valid irrespective of species or any other factor. Is there a dendrochronological study that suggests that these size breakpoints represent pulses of seedlings establishing in the ecotone during periods of favorable weather? Do you have any information on the species composition of these size classes or how the distribution of trees by size class changes by study site? You need to justify/explain your size classes in your methods or in an appendix or supplementary information.

You later mention facilitation and competition, but there is no system-specific context. The reader is left with the vague idea that it might be useful, but without much concrete evidence to demonstrate this. I think that the analysis actually weakens your case that the method is useful, because it is lacking in context and direction or hypothesis. It comes across as being disconnected from the system you're studying.

You also mention in the introduction (L86-88 and L103) and conclusions (L499) that the analysis allows for ecological inference on treeline processes, but you have no data with which to verify your conclusions that competition is the mechanism behind the patterns you observed. You must clarify that your spatial analysis may be useful as a hypothesis generation tool *only*, which is still a useful contribution. That is, you can use these pattern analyses to generate hypotheses about processes, which may then be tested in the field. You must then discuss your specific hypotheses—in the plural—regarding the case study results you present (your point-pattern analysis).

2. thanks for the thoughtful comment. In the Discussion section L380-384 we meant to highlight a limit of the deep learning model which we deemed right to emphasize. We agree with the reviewer that the lack of detection of smaller individuals right in proximity of bigger ones might affect the PPA results. However, as it is also visible in

Figure 3, the detection of small trees was possible and achieved, even when in proximity of big trees. Attached below are a boxplot and violin plot displaying the distribution of omission errors of small trees in function of the distance to medium and tall trees (distance computed as closest distance between the Ground Truth data and the border of the medium/tall crown). As the graphs show, omission errors are uniformly distributed when related to medium trees. When related to tall trees there is a concentration of missed detection on small scales (average value ~9 m), however, these scales greatly surpass the distance over which a small canopy might be obscured or hidden by a bigger one (and also distance values greater that those we might expect to occur between clonal stems, as mentioned in your next comment below) Concerning the discussion of the observed patterns found, L465-469 & L470-484 already discuss possible reasons of the found patterns but we rephrased and implemented chapters 4.1 and 4.3 for better clarity.

**Response:** Lines 381-384 specifically state "In addition to being inherently more challenging to detect in the imagery due to their diminished size, smaller trees often present altered lighting conditions due to being partially obscured or completely concealed by taller ones (Beloiu et al. 2023; Dietenberger et al. 2023; Hamraz et al. 2017), leading to missed detections (i.e., false negatives). The problem is exacerbated in dense clusters (Vauhkonen et al. 2012), common in most of our study sites."

Your figure is potentially convincing that the small crowns are not literally being obscured by larger crowns if the imagery was all taken exactly on-nadir as described in the methods. But, if this is the case, your statement in lines 381-384 is directly contradicted by your own evidence here. So, which do you believe is true? You need to include this figure and discuss the impact of missing the greatest number of small crowns within 5-15 (or even 5-10, looking at the violin plots) meters of tall trees. How would your analysis change if the missed small crowns were included?

I can see, for example, that for most sites, the largest difference between the observed g12(r) and the expected confidence interval is in the range of the distribution of missed small crowns (~ 2-28 m, esp. 5-10 m). It is reasonable to assume that the spatial bias in these missed crowns with respect to taller trees would make the conclusion that the statistically significant relationship between the two size classes is biologically meaningful—with small trees typically found further from large trees, irrespective of species or any other factor—tenuous at best.

The fastest way to resolve this problem is to discuss the very real possibility that the finding is an artifact of missed detections, but that it may be a hypothesis worth investigating in the

future. To be perfectly transparent, you should also discuss this as a limitation of your method. As it stands, you're encouraging readers to draw conclusions from these findings and also encouraging them to replicate your work or use your dataset in the way you've suggested. It is not clear to me that this is defensible.

If you insist on keeping this case study analysis on how the data could be used and you wish to make strong conclusions about the relationship between small and large trees, I would like to see a re-run of the analysis on the gps points of the groundtruthed crowns as opposed to the predicted crowns. Then, you can compare the results and show that the results are not impacted by the bias in missed crowns with proximity to large trees. If you do this, you should still discuss the biological and ecological meaning of your findings in terms of multiple working hypotheses.

3. Thank you for your comment. Regarding the potentially missing small trees, we hope the adjustments described in the previous response address your concerns satisfactorily. As for the definition of individual trees, since our study is focused on the detection of tree canopies and builds upon that, we defined individual trees as "*individual tree crowns clearly separable from the other adjacent crowns* " (definition implemented in chapter 2.2 of the revised manuscript). In a purely remote sensing-based analysis like the one employed in this study, it is not possible to determine whether multiple tree crowns belong to the same individual (e.g., clonal stems), which justifies the definition used.

**Response:** The definition is fine, but this is a limitation you need to discuss. You're assuming that you have clusters of even-aged trees due to competition or facilitation, but this could be because the clusters are actually clusters of clonal stems, which would explain why they are all in the same size class. The same would apply to clusters of stems due to caches made by birds or rodents. You need to state something about the biology of the species you were mapping. You may not be able to say which canopies were which species, but you can at least discuss hypotheses with some actual system-specific context. You should use this context to revise lines 466-475. If this is only a methods paper and you're not prepared to think about your hypotheses with respect to the actual system you are studying, you shouldn't include a system-specific case study to demonstrate the usefulness of your remote sensing methods. Instead, you should focus on the remote sensing methods in this paper and put the application in a separate paper where you can devote more time to thinking through the processes you're speculating on.

Be careful – you treat your size classes as biologically meaningful, but you haven't presented a case for this. They are likely a mix of different species, and growth rates can vary widely in a heterogeneous treeline environment, so they are likely different cohorts

(ages) as well. It therefore doesn't make much sense to discuss "inter-size class competition" or "intra-size class facilitation"—it's not the same as talking about inter- or intra-species competition or facilitation. Furthermore, competition and facilitation are two possibilities, but you're very likely lumping together multiple processes that could be driving these patterns. How, for example, could seed dispersal influence these spatial patterns? If the trees are wind-dispersed, you would expect dispersal to be more abundant downwind of seed sources, and depending on interactions between topography and wind, as well as the typical dispersal distance, you could end up with small seeds flying further from parent trees, followed by strong selective pressure for seedlings to survive to your 50 cm or 130 cm threshold in protected microsites.

You need to discuss multiple hypotheses for the patterns you're seeing, and then you can promote your method as a way to generate hypotheses for processes by analyzing patterns at broader spatial scales. You absolutely cannot claim to draw conclusions about processes using these methods.

It's unclear what the height thresholds are for small, medium, and large, and whether you use the same thresholds for assessing your model performance as for your cluster analysis.

4. we agree with the reviewer that Elliot's work is indeed an example of a similar study that is comparing treeline on an impressively large geographical extent. However, L421-422 *"a study investigating such patterns over large extents across multiple sites simultaneously is unprecedented."* aimed at emphasizing the overall extent of the study areas (90 hectares), not the geographic range. We aimed at highlighting that our study sites are highly representative of the studied landscapes and no previous analysis was performed with such a level of detail on such large extents. We agree with the reviewer and with reviewer#2 that the sentence can be misleading. We hence rephrased as follows: *Several recent studies have highlighted how tree spatial patterns vary along an elevational gradient within the treeline ecotone (Garbarino et al., 2020; Jia et al., 2022; Wang et al., 2021). Other works have investigated tree recruitment at different sites at broad spatial scales (Nicoud et al., 2025), and others investigated spatial patterns on multiple sites in the Pyrenees (Birre et al., 2023). However, to the best of our knowledge, there are no previous studies that have simultaneously investigated the patterns of multiple treelines at the same level of spatial extent (90 ha) and resolution (5cm) as presented in this work.*

**Response:** The revision is mostly acceptable, but if you insist on keeping this statement, you should clarify that this is a remote sensing study vs. a field study. The spatial resolution

doesn't apply to the studies by Elliot and Sindewald, which were in the field. Again, Elliot's spatial extent surpasses yours. I'd like to note that this statement isn't really the strongest part of your work. You're presenting a useful remote sensing method that can support spatial analyses at large geographic extents and small spatial resolutions. Whether or not you were the first to do this exact version of this type of spatial analysis at treeline with this exact spatial resolution and extent isn't as important.

5. Thank for the thoughtful comment. We modified chapter 2.3: *Our methodology consisted of the following steps: i) cropping the RGB orthomosaic of each study site into adjacent tiles of 512 x 512 pixels; ii) systematically selecting 10 tiles per each study site to create the reference dataset; iii) semi-automatic classification of tree crowns; iv) hyperparameter tuning and model calibration using a dataset randomly split into training, validation, and testing subsets; v) performance evaluation; vi) validation of model transferability through spatial cross-validation*. And chapter 2.3.1 was modified and implemented to clarify the aspect: *To generate the training, validation and test datasets, the reference dataset of 100 tiles (512 x 512) was split into 70 % of images for training, 20 % for validation, and 10 % for testing. [···]. The model trained in this way was used to perform predictions on the rest of the tiles to generate tree maps. However, this type of dataset partitioning does not guarantee model transferability since images from all sites are included in each phase of training, validation, and testing. Hence, we performed a spatial cross validation from the beginning to evaluate model generalizability. A k-fold spatial cross-validation was performed using training and validation datasets partitioned according to their geographic distribution. The dataset was partitioned into ten folds based on study sites. In each iteration, images from nine sites were used for training, while the remaining site's images were reserved exclusively for testing. This procedure was repeated across ten iterations, such that each site served as the test set once, thereby ensuring a leave-one-site-out cross-validation scheme*. We also changed *Figure 2* for better clarity, adding an additional panel displaying the model transferability testing as a separate phase to the rest of the workflow. Regarding the use of only 3% of our data for model training, this was not merely a test—this limited subset was indeed the full extent of the training data used, and the reported results reflect the model's performance based solely on that amount.

**Response:** I think I understand better what you did. I was thinking in terms of methods used to train, validate, and test machine learning models during model development including a hyperparameter experiment. If I understand correctly, you did not perform a hyperparameter experiment because you were using a model off-the-shelf. I think you should clarify this fact and describe these two validation methods as two different

experiments. In the first experiment, you tested how well the model would work when trained on only 3% of the data. In the second experiment, you tested how well the model would perform on a geographically independent dataset when trained on other geographically independent datasets. Your work is therefore comprised of two experiments testing the usefulness of this off-the-shelf model for treeline studies in terms of accuracy, as well as a case study applying the model to treeline ecology.

**Follow-up comments:**

1. You need to clarify your groundtruth sampling methods: What height threshold did you use for sampling trees? 50 cm? (Did you ignore all trees smaller than 50 cm? Or did you have a different threshold?)
2. You did record the species of each groundtruthed tree, so what is the species composition of your size classes by study site? What is the joint distribution of species and size class at each study site? Please include this in an appendix or in supplemental information and use this to consider how you're interpreting your PPA findings.
   a. Based on the species distribution by size class, do you see a shift in species composition between size classes at a given site? Or are the classes mixed by species? (Could there be a shift in the species establishing at the site over time due to climate change? Or, perhaps, could different species recruit with greater frequency to different microsites? Could the pulses of recruitment of different species be related to climatic trends?)
   b. If there's no shift in species composition, consider dispersal mechanism and growth patterns. Could you be seeing smaller trees further from the large trees because of high winds at sites dispersing wind-dispersed seeds further away? Could seeds be getting trapped in terrain features that happen to be at a distance from the larger trees? Could animals be caching the seeds in open areas for easy retrieval in the spring because snow melts sooner?  Could each cluster of tree crowns actually be clonal stems of the same individual, each with a separate crown? This last would explain homogeneity in size (and species, if applicable).
3. You need to clarify in your PPA methods whether the size classes defined in lines 247-248 were the same used in your PPA. (You didn't, for example, use the 50 cm breakpoint, correct?) You need to explain these size class breakpoints – why did you choose them? Do they mean the same thing across all study sites given that the study sites have different conditions and sometimes different species? Why did you

ignore the medium trees when you had higher detection accuracy for those? Did you also test those and not find a significant relationship? If that's the case, you should report these results as well. Was the bivariate analysis presented only looking at the relationship between small and tall trees? (Clarify this in the caption of Figure 6.)

4. In line 476, you state that your findings with respect to small and tall trees suggest inter-class competition and are "in line with previous studies findings (Carrer et al. 2013)". Please elaborate on what Carrer et al. found and how this field study supports your conclusion for treeline ecosystems in the Italian Alps.

**Final copyedits:**

1. When you discuss "chapters", you actually mean "sections". The term chapter isn't generally used in this way and applies to sections of a book.
2. Figure 4 should be revised so that you can distinguish the categories when a paper is printed in black-and-white. Please use different symbols for the mean of each class (rather than diamonds for all) to achieve this.
3. In line 352 of the Figure 5 caption, change "black continue line" to "solid black line".